# Occupancy Reward Shaping: Improving Credit Assignment in Offline Goal-Conditioned Reinforcement Learning

**Aravind Venugopal**[*]
Carnegie Mellon University

**Jiayu Chen**
The University of Hong Kong,
INFIFORCE Intelligent Technology

**Xudong Wu**
The University of Hong Kong

**Chongyi Zheng**
Princeton University

**Benjamin Eysenbach**
Princeton University

**Jeff Schneider**
Carnegie Mellon University

## Abstract

The temporal lag between actions and their long-term consequences makes credit assignment a challenge when learning goal-directed behaviors from data. Generative world models capture the distribution of future states an agent may visit, indicating that they have captured temporal information. How can that temporal information be extracted to perform credit assignment? In this paper, we formalize how the temporal information stored in world models encodes the underlying geometry of the world. Leveraging optimal transport, we extract this geometry from a learned model of the occupancy measure into a reward function that captures goal-reaching information. Our resulting method, **Occupancy Reward Shaping (ORS)**, largely mitigates the problem of credit assignment in sparse reward settings. ORS provably does not alter the optimal policy, yet empirically improves performance by $2.2\times$ across 13 diverse long-horizon locomotion and manipulation tasks. Moreover, we demonstrate the effectiveness of ORS in the real world for controlling nuclear fusion on 3 Tokamak control tasks.

Code: https://github.com/aravindvenu7/occupancy_reward_shaping

## 1 Introduction

Reinforcement learning (RL) is fundamentally tied to credit assignment because effectively reasoning about the long-term consequences of actions is crucial to learn diverse goal-reaching behaviors from data. Although goal-conditioned reinforcement learning (GCRL) offers a simple, domain-agnostic and scalable framework for RL from large offline datasets (Liu et al., 2022a; Yang et al., 2023), GCRL agents trained with sparse rewards struggle with credit assignment (Eysenbach et al., 2022; Ding et al., 2019; Wang et al., 2023a; Park et al., 2024a). Therefore, the design of well-defined and meaningful reward functions, called reward shaping (Ng et al., 1999) is a promising solution to address the credit assignment challenge (Yu et al., 2025).

While manual reward design for each goal is infeasible, existing work relies on learning local temporal distance estimators from offline data (Hartikainen et al., 2019) and using them to build semi/non-parametric graphs (Savinov et al., 2018; Mezghani et al., 2023) and indirectly infer temporal information for credit assignment. These approaches run the risk of scaling poorly with task complexity as composing local information to infer global temporal dependencies often leads to compounding errors. So, *can we learn*

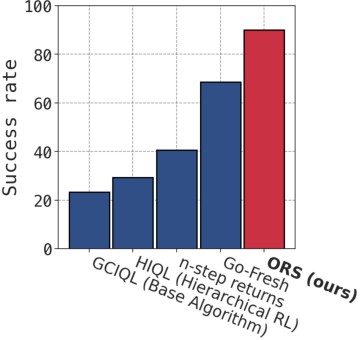

Figure 1: Average relative success rate (task-normalized) on 7 challenging OGBench tasks (Park et al., 2024a). Higher is better; 100.% matches the best algorithm per task. Our method, ORS, performs best, improving over the best baseline by 31%.

---

[1]Corresponding author: avenugo2@andrew.cmu.edu

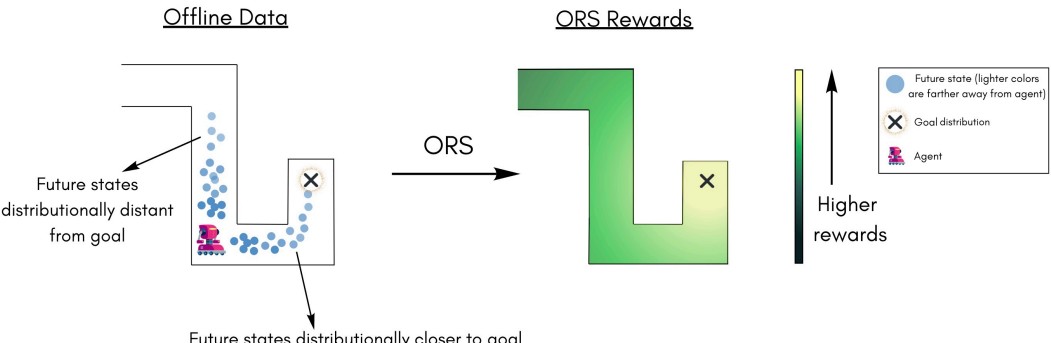

Figure 2: ORS learns the occupancy measure (distribution of future states) and extracts goal-reaching information into a reward function such that states with future states closer to goal have higher rewards.

*reward functions that accurately and directly capture long-term temporal dependencies and goal-reaching information present in offline data?*

Generative world models (Janner et al., 2020b; Schramm & Boularias, 2024; Farebrother et al., 2025) have shown a remarkable ability to capture the multi-modal distribution of future states an agent may visit, i.e., the occupancy measure. This suggests that they must inherently possess the ability to reason about long-term temporal effects of actions by encoding the geometry of the environment. Nonetheless, it remains unclear how to extract such temporal information from a learned world model and leverage it for credit assignment.

In this paper, we formalize how world models encode world geometry. Specifically, using tools for optimal transport, we show that long-term temporal dependencies can be explicitly recovered from the velocity field of a learned occupancy model trained with flow matching (Liu et al., 2022b; Lipman et al., 2022). Building on this insight, we propose **Occupancy Reward Shaping** (ORS), a reward-shaping method that extracts global goal-reaching information from the dataset occupancy measure into a scalar reward while provably preserving the optimal policy. We demonstrate that ORS significantly enhances performance, improving **2.2×** over its sparse-reward base algorithm and achieving **24%-248%** improvements over state-of-the-art methods on 13 challenging locomotion and manipulation tasks from OGBench (Park et al., 2024a). Furthermore, ORS consistently outperforms baselines on 3 tasks that involve controlling actuators in a nuclear fusion reactor called a Tokamak (Char et al., 2023), showing its utility in complex, real-world systems.

## 2 RELATED WORK

**Offline GCRL.** Offline GCRL has been a long-standing area of research in RL (Kaelbling, 1993; Schaul et al., 2015), stemming from a necessity to build innovative algorithms taking advantage of its rich metric structures, probabilistic goal-reaching interpretations and sub-goal compositionality. As a result, diverse types of algorithms exist in offline GCRL based on behavioral cloning (Yang et al., 2022; Hejna et al., 2023), actor-critic methods (Haarnoja et al., 2018; Kostrikov et al., 2021), hind-sight goal relabeling (Andrychowicz et al., 2017) as well as more specialized approaches based on metric learning (Wang et al., 2023a; Reichlin et al., 2024; Park et al., 2024c), contrastive reinforcement learning (Eysenbach et al., 2022), dual RL (Ma et al., 2022; Sikchi et al., 2023b) and hierarchical RL (Chane-Sane et al., 2021; Park et al., 2023; Zhou & Kao, 2025).

Hierarchical RL methods (Chane-Sane et al., 2021; Park et al., 2023) address the challenges posed by long-horizon, sparse-reward tasks with a two-level policy structure where the high level policy predicts sub-goals that a low-level policy learns to reach. Methods such as Ahn et al. (2025) address this challenge by learning an n-step critic (De Asis et al., 2018) to effectively reduce the temporal horizon. Our approach addresses this challenge using reward shaping for effective credit assignment, making ORS complementary to the methods discussed above.

**Reward Shaping.** The idea of using reward shaping to facilitate learning dates back to early RL research and applications (Saksida et al., 1997; Randløv & Alstrøm, 1998). In their canonical work, Ng et al. (1999) proposes potential-based reward shaping (PBRS) that preserves the optimal policy un-

der shaped rewards. Potential functions have been learned using expert demonstrations (Brys et al., 2015; Kang et al., 2018), transitions (Harutyunyan et al., 2015), uncertainty (Li et al., 2025) or with discriminators to be used as exploration bonuses (Durugkar et al., 2021). Another line of research does not follow PBRS and considers reward shaping as a bonus for exploration (Pathak et al., 2017; Mguni et al., 2021; Wang et al., 2023b; Agarwal et al., 2023; Ma et al., 2024; Zheng et al., 2024) and as curriculum learning (Andrychowicz et al., 2017; Eysenbach et al., 2022; Zheng et al., 2023).

Hartikainen et al. (2019) estimates local temporal distance to provide a dense reward for credit assignment in online GCRL. We address the challenge of learning shaped rewards for credit assignment in Offline GCRL. The closest work to ours, Mezghani et al. (2023), uses both a local reward computed with a local temporal distance classifier and a "global" reward computed using shortest path search on a graph constructed using the local distance classifier. While graph-based methods scale poorly with task complexity, we learn a single reward function capable of directly encoding global, long-horizon goal-reaching information for credit assignment by leveraging the occupancy measure. Our experiments show the effectiveness of ORS over prior state-of-the-art work.

## 3 BACKGROUND

### 3.1 NOTATION

**Goal-Conditioned Reinforcement Learning:** We consider an infinite-horizon controlled Markov Process (a MDP (Puterman, 2014) without rewards) defined by $(\mathcal{S}, \mathcal{A}, \mu, p, \gamma)$ with state space $\mathcal{S}$ and action space $\mathcal{A}$. $p: S \times A \times S \to \mathbb{R}$ denotes the transition function while $\mu$ denotes the initial state distribution. The discount factor is denoted by $\gamma \in (0, 1)$. A policy $\pi(a|s): S \times A \to \mathbb{R}$ is a probability distribution mapping states to actions. Each policy induces a conditional state-action occupancy distribution $d^\pi(s^+ \mid s, a)$ over future states $s^+$: $d^\pi(s^+ \mid s, a) = (1 - \gamma) \sum_{\Delta t=1}^\infty \gamma^{\Delta t-1} \mathbb{P}(s_{t+\Delta t} = s^+ \mid s, a, \pi)$, which we refer to as the occupancy measure for simplicity.

The objective of GCRL is to reach any goal, $g \in \mathcal{S}$, from another state $s \in \mathcal{S}$ in the minimum number of steps. A goal-conditioned policy $\pi(a|s, g): S \times S \times A \to \mathbb{R}$ maps states and goals to actions. Formally, the aim is to learn the optimal shortest-path policy $\pi^*(a|s, g)$ that maximizes $\mathbb{E}_{\tau \sim p(\tau|g)} \sum_{t=0}^\infty \gamma^t \delta_g(s_t)$ where $t$ is the timestep, $p(\tau \mid g) = \mu(s_0)\pi(a_0 \mid s_0, g)p(s_1 \mid s_0, a_0) \cdots p(s_i \mid s_{i-1}, a_{i-1}) \cdots$ and $\delta_g$ is a Dirac delta at goal $g$. Such a policy reaches $g$ in the minimum number of steps. In this paper, we focus on offline GCRL. In particular, agents cannot interact with the environment for learning but have access to an offline dataset of trajectories $\mathcal{D}$, where each trajectory is $\tau = (s_0, a_0, s_1, a_1, \ldots, ;, s_T)$. We denote the dataset behavioral policy by $\pi_\mathcal{D}(a \mid s)$ and the dataset occupancy measure corresponding to $\pi_\mathcal{D}$ by $d^{\pi_\mathcal{D}}(s^+ \mid s, a)$.

**Flow Matching:** Flow matching aims to learn to sample from a distribution $p_{\mathbf{data}}$, given a finite number of samples $x^{(1)}, ..., x^{(N)} \in \mathbf{R}^d$ drawn from $p_{\mathbf{data}}$. Flow matching transforms a base distribution $p_0$, which is typically a simple Isotropic Gaussian $\mathcal{N}(0, \mathbf{I_d})$ at time $t = 0$ to the target data distribution $p_{\mathbf{data}}(x)$ at $t = 1$. This transformation is defined by a velocity field $v_\theta(t, x): [0, 1] \times \mathbf{R}^d \to \mathbf{R}^d$, parameterized by a neural network and having a corresponding flow $\psi_\theta(t, x): [0, 1] \times \mathbf{R}^d \to \mathbf{R}^d$ which is a unique solution to the Ordinary Differential Equation (ODE) $\frac{d}{dt}\psi_\theta(t, x) = v_\theta(t, \psi_\theta(t, x))$. The velocity field is trained to minimize:

$$\min_\theta \ \mathbb{E}_{x_0 \sim \mathcal{N}(0, I_d), \ x_1 \sim p(x), \ t \sim \text{Unif}(0,1)} \big\| v_\theta(t, x_t) - (x_1 - x_0) \big\|_2^2 \tag{1}$$

where $\text{Unif}([0, 1])$ denotes uniform sampling between 0 and 1 and $x_t = (1 - t) x_0 + t x_1$ is the linear interpolation between $x_0$ and $x_1$. On convergence, $v_\theta(t, x_t)$ learns the velocity field which transforms samples from $p_0$ to samples from $p_{\mathbf{data}}$ by numerically integrating the ODE. In this paper, following Lipman et al. (2024); Park et al. (2025), we use linear interpolation between base and target distributions and uniform time sampling. As in Park et al. (2025), we find that the Euler method is sufficient for solving the ODE.

### 3.2 MOTIVATION

In this section, we motivate our method by examining why credit assignment becomes a challenge when using sparse rewards. We do so by analyzing the learning dynamics of the goal-conditioned

value function $V(s, g)$ the GCRL policy $\pi(a|s, g)$ must maximize. Given a goal g, initial state $s_0$, the optimal policy $\pi^*(a^*|s, g)$, and the optimal trajectory $\tau^* = \{s_0, s_1, ..., s_T = g\}$ induced by $\pi^*$, the optimal value function is monotonically non-decreasing along $\tau^*$: $\forall s_i, s_j \in \tau^*, V^*(s_j, g) \geq V^*(s_i, g) \iff j \geq i$. However, non-monotonicity often arises in practice due to sampling or approximation errors (Park et al., 2023; Ahn et al., 2025).

High non-monotonicity in $\hat{V}(s, g)$ can cause the policy to get stuck in sub-optimal regions of the state space, especially when $g$ is distant from $s$. We hypothesize that in long-horizon tasks, under sparse rewards, credit assignment becomes a challenge because $\hat{V}(s, g)$ exhibits high non-monotonicity, leading to poor policies.

We now evaluate this hypothesis empirically using expert trajectories, each with a different length and a unique goal, on the sparse-reward **antmaze-giant-navigate** task from OGBench (Park et al., 2024a) used in Ahn et al. (2025). Trajectories are visualized in Appendix B.4. We analytically compute $\hat{V}(s, g)$ as the discounted sum of rewards. By injecting multiplicative noise with variance $\sigma_v^2$, we simulate errors in value estimation: $\hat{V}(s, g) \leftarrow r(s, a^*, g) + \gamma * (\hat{V}(s', g) + \epsilon * \hat{V}(s', g)), \epsilon \sim \mathcal{N}(0, \sigma_v^2)$, with $\gamma = 0.99$. We evaluate the quality of $\hat{V}(s', g)$ using $\delta_V$, the value non-monotonicity error, computed as the proportion of states along an expert trajectory where $\hat{V}(s_{t+1}, g) < \hat{V}(s_t, g)$.

Figure 3 plots average $\delta_V$ over varying $\sigma_v^2$ while Fig. 4 plots $\hat{V}(s, g)$ over 5 trajectories for $\sigma_v = 0.0005$. The results indicate that under sparse rewards, $\delta_V$ is high even under small $\sigma_v$ and increases with $\sigma_v$ and task horizon. This motivates the central question we aim to address:

*How do we design reward functions that encode long-horizon temporal structure and thereby mitigate value non-monotonicity with effective credit assignment, leading to performative policies?*

Figure 3: Average $\delta_V$ (**y-axis**) over 5 expert trajectories on **antmaze-giant-navigate** as a function of $\sigma_v$ (**x-axis**). $\delta_V$ increases with $\sigma_v$, denoting increasingly more noisy and non-monotonic $\hat{V}(s, g)$ under sparse rewards.

## 4 OCCUPANCY REWARD SHAPING FOR OFFLINE GCRL

First, we detail how we learn a generative model of the occupancy measure (Sec. 4.1). In Sec.4.2, we then show exactly how the occupancy measure encodes state geometry (Prop.1). This allows us to extract goal-reaching temporal information from the learned occupancy model into a reward-shaping term (Prop.2). We show that these "occupancy-shaped" rewards preserve the optimal goal-reaching policy (Sec.4.3) and finally, we summarize our method in Sec.4.4.

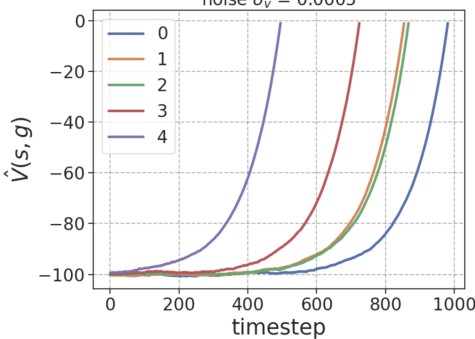

### 4.1 LEARNING THE OCCUPANCY MEASURE

The state-action conditioned occupancy measure over future states of the dataset policy $\pi_\mathcal{D}$ has the following recursive form (Sutton, 1995; Janner et al., 2020a; Schramm & Boularias, 2024):

Figure 4: $\hat{V}(s, g)$ (**y-axis**) with sparse rewards vs time (**x-axis**) over 5 expert trajectories (each line a trajectory) from **antmaze-giant-navigate**. $\hat{V}(s, g)$ becomes increasingly noisy as we move away from goal towards the start of a trajectory (moving left on x-axis) even under small $\sigma_v$.

$$d_\theta^{\pi_\mathcal{D}}(s^+ \mid s, a) = (1 - \gamma)\, p(s' \mid s, a) + \gamma\, d_{\theta_-}^{\pi_\mathcal{D}}(s^+ \mid s', a'), \qquad \forall(s, a, s', a') \in \mathcal{D} \qquad (2)$$

This recursive form, reminiscent of temporal difference learning, allows learning $d^{\pi_\mathcal{D}}(s^+ \mid s, a)$ parameterized by $\theta$ as a mixture over two components: **1.** the transition function and **2.** a target

$d^{\pi_\mathcal{D}}(s^+ \mid s', a')$ over subsequent future states under $\pi_\mathcal{D}$ (parameterized by $\theta_-$, a time-delayed version of $\theta$). This approach naturally stitches together future states under intersecting trajectories in $\mathcal{D}$. To accurately represent the multi-modality inherent in $d^{\pi_\mathcal{D}}$ and to enable efficient computation of the ORS reward function (explained in Sec. 4.2), we learn $d^{\pi_\mathcal{D}}$ using a flow matching model with the following loss (Farebrother et al., 2025) :

$$\mathcal{L}_{\text{flow}}(\theta) = (1 - \gamma)\,\mathcal{L}_{\text{next}}(\theta) + \gamma\,\mathcal{L}_{\text{future}}(\theta); \qquad \forall\, (s, a, s', a') \in \mathcal{D} \tag{3}$$

$$\mathcal{L}_{\text{next}}(\theta) = \mathbb{E}_{\substack{s'=x_1\sim\mathcal{D}\\x_0\sim\mathcal{N}(0, I_d)\\t\sim\text{Unif}([0,1])}} \left[\left\|v_\theta(t, s, a, x_t) - (x_1 - x_0)\right\|_2^2\right];$$

$$\mathcal{L}_{\text{future}}(\theta) = \mathbb{E}_{\substack{s^+=x_1\sim d_{\theta_-}^{\pi_\mathcal{D}}(.|s',a')\\x_0\sim\mathcal{N}(0, I_d)\\t\sim\text{Unif}([0,1])}} \left[\left\|v_\theta(t, s, a, x_t) - \mathbf{sg}[v_{\theta_-}(t, s', a', x_t)]\right\|_2^2\right].$$

where $\mathcal{L}_{next}(\theta)$ is the standard flow matching loss over the transition function and $\mathcal{L}_{future}(\theta)$ regresses the velocity field $v_\theta(t, s, a, x_t)$ towards a bootstrapped estimate of the velocity field of $d_{\theta_-}^{\pi_\mathcal{D}}(s^+|s', a')$ represented by $v_{\theta_-}(t, s', a', x_t)$. **sg** represents the stop-gradient operation.

## 4.2 EXTRACTING TEMPORAL INFORMATION FROM THE OCCUPANCY MEASURE

We now establish a precise connection between the occupancy measure and state space geometry.

**Definition 1** (Shortest-path distance and level sets). *Let $step^*(s, g)$ denote the minimum number of steps required to reach goal $g$ from any state $s$ ; $\forall (s, g) \in \mathcal{D}$. We define the level sets:*

$$\mathcal{S}_k = \{s \in \mathcal{S} : step^*(s, g) = k\}, \quad k = 0, 1, 2, \ldots$$

*Clearly, $\mathcal{S}_0 = \{g\}$. Under deterministic dynamics, for any $s \in \mathcal{S}_k$ with $k \geq 1$, there exists an action $a^*$ at state $s$ such that the next-state distribution satisfies $p(\cdot \mid s, a^*) \subseteq \mathcal{S}_{k-1}$.*

This characterization of distance from goal allows us to quantify how the occupancy measure encodes geometry in terms of the squared *Wasserstein-2 distance* between $d^{\pi_\mathcal{D}}$ and $\delta_g$:

**Proposition 1** (informal). *Under suitable assumptions on goal reachability, dynamics, and dataset coverage, for all $(s, a, g) \in \mathcal{D}$, the average squared Wasserstein-2 distance*

$$W_2^2\big(\delta_g, d^{\pi_\mathcal{D}}(s^+ \mid s)\big) := \mathbb{E}_{a\sim\pi_\mathcal{D}(\cdot|s)}\Big[W_2^2\big(\delta_g, d^{\pi_\mathcal{D}}(s^+ \mid s, a)\big)\Big]$$

*is monotonically increasing with respect to the shortest-path layer index $k$. In particular, for a fixed goal $g$ and any $k \geq 1$, if $s_1 \in \mathcal{S}_{k-1}$ and $s_2 \in \mathcal{S}_k$, then*

$$W_2^2\big(\delta_g, d^{\pi_\mathcal{D}}(s^+ \mid s_1)\big) \; \leq \; W_2^2\big(\delta_g, d^{\pi_\mathcal{D}}(s^+ \mid s_2)\big).$$

*Moreover, for any state–goal pair $(s, g)$, the squared Wasserstein-2 distance is minimized by the optimal action:*

$$W_2^2\big(\delta_g, d^{\pi_\mathcal{D}}(s^+ \mid s, a^*)\big) \; \leq \; W_2^2\big(\delta_g, d^{\pi_\mathcal{D}}(s^+ \mid s, a)\big), \quad \forall a \in \mathcal{A}, \; a^* \sim \pi^*(\cdot \mid s, g).$$

The assumptions and proof are provided in Appendix A.3 and Appendix A.4. This formulation captures not only *how far* the center of mass of $d^{\pi_\mathcal{D}}(s^+ \mid s, a)$ is from $g$, but also *how spread out* $d^{\pi_\mathcal{D}}(s^+ \mid s, a)$) is with respect to $g$: if state $A$ and state $B$ are both 5 transitions away from $g$ but $A$ has more future states that are farther away from the goal than $B$, $W_2^2(;)$ will be lower for state $B$.

Intuitively, this means that by defining a reward $r^W(s, a, g) = -W_2^2(\delta_g, d^{\pi_\mathcal{D}}(s^+ \mid s, a))$, we can extract the state-space geometry as goal-reaching temporal information into a scalar value that is higher for states with $s^+$ closer to $g$ under $\mathcal{D}$. Unlike prior works (Mezghani et al., 2023; Hartikainen et al., 2019), this reward-shaping term will directly capture global long-horizon information about goal-reachability. What remains is to compute $W_2^2(\delta_g, d^{\pi_\mathcal{D}}(s^+ \mid s, a))$. While exact computation involves an intractable integral, we show how to estimate $r^W(s, a, g)$ using the velocity field corresponding to $d_\theta^{\pi_\mathcal{D}}(s^+ \mid s, a)$:

---

**Algorithm 1:** Occupancy Reward Shaping.

---

**Input:** Occupancy model $d_\theta^{\pi_\mathcal{D}}$, reward function $r_\psi^W$, learning rate $\eta$

    // Training the occupancy model:
1 **for** *each iteration* $t = 1$ *to* $N$ **do**
2     Sample $(s, a, s^+) \sim \mathcal{D}$;
3     Compute $\mathcal{L}_{pretrain}(\theta)$ (Eq. 8) and update $\theta \leftarrow \theta - \eta \nabla_\theta \mathcal{L}_{pretrain}(\theta)$;
4 **end**
5 **for** *each iteration* $t = 1$ *to* $N$ **do**
6     Sample $(s, a, s', a') \sim \mathcal{D}$;
7     Compute $\mathcal{L}_{flow}(\theta)$ (Eq. 3) and update $\theta \leftarrow \theta - \eta \nabla_\theta \mathcal{L}_{flow}(\theta)$;
8 **end**
    // Training the reward function:
9 **for** *each iteration* $t = 1$ *to* $N$ **do**
10     Sample $(s, a, g) \sim \mathcal{D}$;
11     Compute $\mathcal{L}_{rew}(\psi)$ (Eq. 5) and update $\psi \leftarrow \psi - \eta \nabla_\psi \mathcal{L}_{rew}(\psi)$;
12 **end**

---

**Proposition 2.** *The squared Wasserstein-2 distance between $\delta_g$ and $d^{\pi_\mathcal{D}}(s^+ \mid s, a)$ is upper bounded, up to a multiplicative constant $C > 0$, by the mean squared error of the associated velocity field corresponding to $d^{\pi_\mathcal{D}}(s^+ \mid s, a)$ and the target velocity corresponding to $\delta_g$:*

$$W_2^2\left(\delta_g, d^{\pi_\mathcal{D}}(s^+ \mid s, a)\right) \leq C \mathop{\mathbb{E}}_{\substack{x_1 \sim \delta_g \\ x_0 \sim \mathcal{N}(0, I_d) \\ t \sim \mathrm{Unif}([0,1])}} \left\| v(t, s, a, x_t) - (x_1 - x_0) \right\|_2^2, \quad (4)$$

*where $x_t = tx_1 + (1-t)x_0$.*

where $v(t, s, a, x_t)$ is the velocity field corresponding to $d^{\pi_\mathcal{D}}(s^+|s, a)$ at time $t$. The proof is in Appendix A.5. As mentioned in Sec. 4.1, this further justifies our choice of using a flow matching model to learn $d^{\pi_\mathcal{D}}$. Using the learned $d_\theta^{\pi_\mathcal{D}}$ from Sec. 4.1, we can now learn $r^W(s, a, g)$ using a neural network $\psi$. Notably, this does not involve running an ODE solver for multiple time-steps:

$$\mathcal{L}_{\mathrm{rew}}(\psi) = \mathbb{E}_{s,a,g \sim \mathcal{D}} \left[ \left\| \hat{r^W}_\psi(s, a, g) - \left[ -\mathop{\mathbb{E}}_{\substack{x_1 = g \\ x_0 \sim \mathcal{N}(0, I_d) \\ t \sim \mathrm{Unif}([0,1])}} \left\| v_\theta(t, s, a, x_t) - (x_1 - x_0) \right\|_2^2 \right] \right\|_2^2 \right] \quad (5)$$

### 4.3 ORS preserves the optimal policy

It remains unclear whether using $r_W$ from Sec.4.2, one can learn the same optimal policy as with sparse rewards, as it is estimated using the *dataset occupancy measure* $d^{\pi_\mathcal{D}}$.

**Theorem 1.** *Under suitable assumptions on goal reachability, dynamics, and dataset coverage, the greedy goal-conditioned policy:*

$$\pi^{\mathrm{greedy}}(a \mid s, g) = \arg\max_{a \in \mathcal{A}} Q^*(s, a, g),$$

*where $Q^*(s, a, g)$ denotes the optimal action-value function induced by the reward $r^W(s, a, g)$, coincides with the optimal shortest-path policy $\pi^*(a \mid s, g)$.*

Please refer to Appendix A.6 for the proof. ORS thus enables efficient use of the rich goal-reaching information present in the dataset occupancy measure without having to estimate the new occupancy measure at each intermediate policy improvement step.

### 4.4 Method Summary

ORS has 3 stages: **1.** Train the flow matching occupancy model $d_\theta^{\pi_\mathcal{D}}$ using Eq. 3; **2.** Train the reward function $r_\psi^W$ using Eq. 5; and **3.** Train a goal-conditioned policy using any offline GCRL algorithm that uses a TD-learning critic. Stages 1 and 2 that involve learning $d_\theta^{\pi_\mathcal{D}}$ and $r_\psi^W$ are summarized in Alg. 1. Details on architecture and hyperparameters are provided in Appendix B.6.

## 5 EXPERIMENTS

In this section, we present an extensive empirical analysis of ORS across a variety of challenging long-horizon offline GCRL tasks. We then conduct detailed analyses and ablations to dissect key design choices underlying our algorithm.

***Tasks:*** For our empirical analysis, we primarily use OGBench (Park et al., 2024a), a benchmark specifically built for evaluating offline GCRL algorithms. We choose OGBench over older benchmarks such as Fu et al. (2020); Tarasov et al. (2023) for its task diversity, challenging long-horizon sparse-reward tasks that are unsaturated as of 2025, and comprehensive multi-goal evaluation.

OGBench tasks are broadly categorized into 4 types: maze navigation, cube manipulation, puzzle manipulation, and scene tasks. We select tasks with varying levels of task complexity and planning horizon. from OGBench to be representative of these 4 categories. Within each category, we include tasks of varying complexity (e.g., cube-double and cube-triple) and varying dataset quality (e.g., cube-triple-play and cube-triple-noisy). We provide a detailed explanation of tasks in Appendix B.1.

To demonstrate effectiveness in real-world settings, we evaluate ORS on actuator control tasks in a nuclear fusion reactor called a Tokamak. We use raw sensor and actuator data collected from the DIII-D tokamak located in San Diego, CA, USA. ORS controls 4 actuators that influence evolution of plasma in the Tokamak, a system characterized by highly non-linear, stochastic dynamics and complex physics. We evaluate over 3 separate tasks, each tracking a certain scalar or profile state variable: $\beta_N$ (normalized ratio between plasma pressure and magnetic pressure), Electron density and Ion rotation. These quantities critically influence plasma stability and overall power output required for commercially viable nuclear fusion. Please refer to Appendix B.2 for further details.

***Baselines:*** We compare ORS against a set of representative baselines. These include Goal Conditioned Behavioral Cloning (**GCBC**) (Ghosh et al., 2019); Goal-Conditioned Implicit Q Learning (**GCIQL**) (Kostrikov et al., 2021) and Goal-Conditioned Implicit Value Learning (**GCIVL**) (Park et al., 2023) which approximate value functions using expectile regression (Newey & Powell, 1987) and recover policies using behavior-constrained deterministic policy gradient (DDPG + BC) and advantage-weighted regression (AWR), respectively (Park et al., 2024b). Quasimetric RL (**QRL**) (Wang et al., 2023a) learns a specialized quasimetric goal-conditioned value function with a dual objective. Contrastive RL (**CRL**) (Eysenbach et al., 2022) fits a Monte Carlo goal-conditioned value function using contrastive learning and extracts a greedy policy from it.

We also compare ORS to several methods designed for long-horizon, sparse-reward settings. **Go-Fresh** (Mezghani et al., 2023) learns a shaped reward as a sum of a local reward from a temporal distance classifier between states and a global reward computed by shortest-path search on a graph constructed with the local distance classifier. **SMORE** (Sikchi et al., 2023a) is an occupancy-matching method that uses dual RL to learn unnormalized densities that reflect the distance to goal.

Hierarchical Implicit Q Learning (**HIQL**) (Park et al., 2023) trains a low-level policy to reach goals sampled from a high-level policy, both trained via a single GCIVL-style value function. Subgoal Advantage-Weighted Policy Bootstrapping (**SAW**) (Zhou & Kao, 2025) improves over HIQL by training a non-hierarchical goal-conditioned policy by bootstrapping on subgoal-conditioned policies with advantage-weighted importance sampling. **n-step GCIQL** and **GCIQL-OTA** use different techniques to learn a GCIQL critic using n-step TD learning (De Asis et al., 2018; Ahn et al., 2025). These baselines ensure a standalone comparison of our method against two key strategies to horizon reduction in RL. Both ORS and Go-Fresh use GCIQL with a Gaussian policy, owing to its popularity and simplicity and to ensure fair comparison. We build on the codebase of OGBench and make use of the official implementation for Go-Fresh. All algorithms use hindsight relabeling (Kaelbling, 1993; Andrychowicz et al., 2017). Appendix B.3-B.8 provides additional information on baselines, hyperparameters used and wall-clock time.

### 5.1 RESULTS

Performance on OGBench is measured by average (binary) success rates on 5 test-time goals of each task. We train algorithms to convergence and average the results over **8 seeds**. We list the number of training iterations per task in Appendix B.5. Performance on Tokamak tasks is measured as the

Table 1: ORS vs baselines: Overall average (binary) success rate (%) across the 5 testtime goals over 8 seeds per task per algorithm. We report the 95% bootstrapped C.I. size after the ± sign.

| Dataset | GCBC | GC-IVL | QRL | CRL | GC-IQL | Go-Fresh | ORS (ours) |
|---|---|---|---|---|---|---|---|
| antmaze-large-navigate | $25 \pm 3$ | $18 \pm 3$ | $64 \pm 18$ | $\mathbf{90 \pm 4}$ | $34 \pm 4$ | $\mathbf{88 \pm 3}$ | $\mathbf{88 \pm 7}$ |
| antmaze-giant-navigate | $0 \pm 0$ | $0 \pm 0$ | $9 \pm 4$ | $39 \pm 8$ | $0 \pm 0$ | $30 \pm 10$ | $\mathbf{56 \pm 9}$ |
| cube-double-play | $1 \pm 1$ | $36 \pm 3$ | $1 \pm 0$ | $10 \pm 2$ | $\mathbf{40 \pm 5}$ | $17 \pm 6$ | $\mathbf{45 \pm 7}$ |
| cube-triple-play | $0 \pm 0$ | $1 \pm 1$ | $0 \pm 0$ | $6 \pm 3$ | $7 \pm 3$ | $18 \pm 5$ | $\mathbf{37 \pm 8}$ |
| puzzle-4x4-play | $0 \pm 0$ | $13 \pm 2$ | $0 \pm 0$ | $0 \pm 0$ | $26 \pm 3$ | $\mathbf{74 \pm 6}$ | $70 \pm 5$ |
| puzzle-4x5-play | $0 \pm 0$ | $7 \pm 1$ | $0 \pm 0$ | $1 \pm 0$ | $14 \pm 1$ | $\mathbf{20 \pm 1}$ | $\mathbf{20 \pm 0}$ |
| puzzle-4x6-play | $0 \pm 0$ | $10 \pm 2$ | $0 \pm 0$ | $4 \pm 1$ | $12 \pm 1$ | $17 \pm 4$ | $\mathbf{20 \pm 2}$ |
| scene-play | $5 \pm 1$ | $42 \pm 4$ | $5 \pm 1$ | $19 \pm 2$ | $51 \pm 4$ | $56 \pm 10$ | $\mathbf{80 \pm 4}$ |
| antmaze-large-explore | $0 \pm 0$ | $8 \pm 6$ | $0 \pm 0$ | $0 \pm 0$ | $1 \pm 1$ | $\mathbf{38 \pm 10}$ | $22 \pm 7$ |
| cube-triple-noisy | $1 \pm 1$ | $9 \pm 1$ | $1 \pm 0$ | $3 \pm 1$ | $2 \pm 1$ | $5 \pm 4$ | $\mathbf{22 \pm 7}$ |
| puzzle-4x4-noisy | $0 \pm 0$ | $20 \pm 3$ | $0 \pm 0$ | $0 \pm 0$ | $29 \pm 7$ | $50 \pm 5$ | $\mathbf{56 \pm 7}$ |
| puzzle-4x6-noisy | $0 \pm 0$ | $17 \pm 2$ | $0 \pm 0$ | $6 \pm 3$ | $\mathbf{18 \pm 2}$ | $\mathbf{19 \pm 4}$ | $\mathbf{19 \pm 1}$ |
| scene-noisy | $1 \pm 1$ | $26 \pm 5$ | $9 \pm 1$ | $1 \pm 1$ | $26 \pm 2$ | $34 \pm 5$ | $\mathbf{40 \pm 5}$ |
| **Mean** | 2.5 | 15.9 | 6.9 | 13.7 | 20.0 | 35.8 | **44.2** |

cumulative reward per episode, where reward is the L2 distance between the goal variables to track and the actual achieved quantities. On Tokamak tasks, we evaluate on 10 test-time goals over **4 seeds**. We discuss the results below:

***How effective is ORS?*** ORS achieves the best performance on most tasks, with especially large gains on more complex domains. Table 1 summarizes results over 12 offline locomotion and manipulation datasets, where **\*-play/\*-navigate** denote noisy expert datasets and **\*-noisy/\*-explore** denote highly sub-optimal datasets (Park et al., 2024a). ORS achieves a $\mathbf{2.2\times}$ **improvement in performance on average** over its base algorithm GCIQL that uses sparse rewards. While we see that GCIQL/GCIVL struggle on locomotion and QRL/CRL struggle on manipulation, ORS consistently outperforms these baselines on both domains. ORS outperforms Go-Fresh on the majority of tasks, demonstrating the effectiveness of occupancy-based reward shaping over graph-based reward shaping, especially in complex tasks such as **antmaze-giant-navigate** and **cube-triple-play** that are characterized by long horizons upto 2000 steps long (Park et al., 2024a). We also see that ORS remains effective even under highly sub-optimal data in the **\*-noisy/\*-explore** datasets. Notably, only ORS and Go-Fresh produce non-zero performance on **antmaze-large-explore**. Although Go-Fresh outperforms ORS on this task, we hypothesize that this could be because the additional local rewards used by Go-Fresh are effective in such a task.

***How does ORS compare to other strategies for long-horizon, sparse-reward offline GCRL?*** On average, ORS outperforms the next-best method GO-Fresh by 24% and achieves $2.2\times$ and $1.9\times$ high performance than hierarchical (HIQL) and n-step return approaches(GCIQL-OTA), respectively. From Table 2, we see that while hierarchical or policy-bootstrapped-based methods like

Table 2: ORS vs long-horizon sparse-reward offline GCRL strategies on OGBench tasks: Overall average (binary) success rate (%) across the 5 test time goals over 8 seeds per task per algorithm. We report the 95% bootstrapped C.I. size after the ± sign.

| Dataset | HIQL | SAW | SMORE | n-step GCIQL | GCIQL-OTA | Go-Fresh | ORS (ours) |
|---|---|---|---|---|---|---|---|
| antmaze-large-navigate | $\mathbf{91 \pm 2}$ | $86 \pm 5$ | $22 \pm 5$ | $53 \pm 9$ | $90 \pm 4$ | $88 \pm 3$ | $88 \pm 7$ |
| antmaze-giant-navigate | $\mathbf{72 \pm 7}$ | $48 \pm 9$ | $1 \pm 1$ | $1 \pm 1$ | $26 \pm 5$ | $30 \pm 10$ | $56 \pm 9$ |
| cube-double-play | $6 \pm 2$ | $\mathbf{40 \pm 7}$ | $2 \pm 2$ | $4 \pm 3$ | $3 \pm 2$ | $17 \pm 6$ | $\mathbf{45 \pm 7}$ |
| cube-triple-play | $3 \pm 2$ | $6 \pm 6$ | $0 \pm 0$ | $1 \pm 1$ | $2 \pm 2$ | $18 \pm 5$ | $\mathbf{37 \pm 8}$ |
| puzzle-4x4-play | $7 \pm 2$ | $17 \pm 12$ | $0 \pm 0$ | $46 \pm 5$ | $\mathbf{85 \pm 4}$ | $74 \pm 6$ | $70 \pm 5$ |
| puzzle-4x5-play | $8 \pm 4$ | $8 \pm 4$ | $0 \pm 0$ | $5 \pm 2$ | $19 \pm 1$ | $\mathbf{20 \pm 1}$ | $\mathbf{20 \pm 0}$ |
| puzzle-4x6-play | $3 \pm 1$ | $8 \pm 4$ | $0 \pm 0$ | $14 \pm 3$ | $15 \pm 3$ | $17 \pm 4$ | $\mathbf{20 \pm 2}$ |
| scene-play | $38 \pm 3$ | $63 \pm 6$ | $8 \pm 2$ | $26 \pm 7$ | $42 \pm 7$ | $56 \pm 10$ | $\mathbf{80 \pm 4}$ |
| antmaze-large-explore | $0 \pm 0$ | $15 \pm 8$ | $0 \pm 0$ | $0 \pm 0$ | $0 \pm 0$ | $\mathbf{38 \pm 10}$ | $22 \pm 7$ |
| cube-triple-noisy | $2 \pm 1$ | $0 \pm 0$ | $1 \pm 1$ | $2 \pm 1$ | $2 \pm 1$ | $5 \pm 4$ | $\mathbf{22 \pm 7}$ |
| puzzle-4x4-noisy | $3 \pm 3$ | $3 \pm 3$ | $0 \pm 0$ | $0 \pm 0$ | $0 \pm 0$ | $50 \pm 5$ | $\mathbf{56 \pm 7}$ |
| puzzle-4x6-noisy | $2 \pm 1$ | $8 \pm 6$ | $0 \pm 0$ | $12 \pm 6$ | $15 \pm 3$ | $\mathbf{19 \pm 4}$ | $\mathbf{19 \pm 1}$ |
| scene-noisy | $25 \pm 4$ | $33 \pm 6$ | $3 \pm 2$ | $2 \pm 2$ | $3 \pm 2$ | $34 \pm 5$ | $\mathbf{40 \pm 5}$ |
| **Mean** | 20.0 | 25.8 | 2.8 | 12.7 | 23.2 | 35.8 | **44.2** |

Table 3: ORS vs long-horizon sparse-reward offline GCRL strategies on 3 challenging Tokamak control tasks: Overall average cumulative return across 10 testtime goals over 4 seeds per task per algorithm. We report the 95% bootstrapped C.I. size after the ± sign.

| Task | HIQL | SAW | SMORE | n-step GCIQL | GCIQL-OTA | GO-Fresh | ORS |
|---|---|---|---|---|---|---|---|
| Tokamak $\beta_N$ | **-45.26 ± 3.83** | **-51.39 ± 8.16** | $-56.06 \pm 12.71$ | $-88.7 \pm 44.3$ | $-67.73 \pm 9.73$ | **-48.20 ± 11.72** | **-44.76 ± 8.42** |
| Tokamak density | $-49.8 \pm 7.5$ | $-44.0 \pm 2.3$ | $-54.0 \pm 10.8$ | **-25.6 ± 4.1** | **-27.1 ± 2.5** | $-38.6 \pm 12.6$ | **-26.9 ± 6.6** |
| Tokamak rotation | $-24.4 \pm 1.3$ | $-26.4 \pm 5.1$ | $-28.5 \pm 6.7$ | $-57.4 \pm 46.6$ | $-30.1 \pm 6.9$ | $-28.9 \pm 7.4$ | **-22.4 ± 0.6** |

HIQL and SAW are effective on antmaze locomotion but struggle on manipulation tasks. SMORE is largely ineffective with a mean performance of just 2.8% while n-step return methods perform poorly on noisy datasets. In contrast, the strong performance of ORS with a simple non-hierarchical policy underscores the effectiveness of its occupancy-based reward shaping.

***Is ORS effective in real-world tasks?*** We now examine the ability of ORS to track goals by controlling actuators in real-world Tokamak control tasks. Table 3 compares ORS with baselines specifically designed for long-horizon, sparse-reward offline GCRL. While HIQL and SAW perform relatively well for rotation control, they perform poorly on other tasks. While SMORE performs poorly overall, both n-step GCIQL and GCIQL-OTA perform well on density control but show poor performance on other tasks. GO-Fresh performs poorly overall on all tasks; we believe this is because graph-based reward shaping is ineffective in tasks with stochastic dynamics. ORS consistently achieves best performance across the Tokamak control, highlighting its effectiveness in complex real-world tasks.

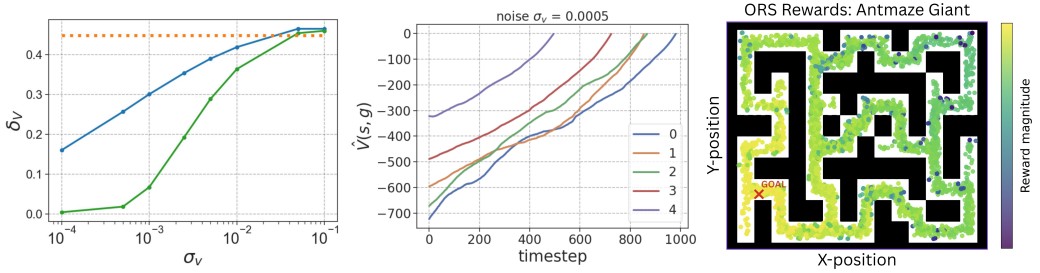

Figure 5: **Left:** ORS leads to *lower average value non-monotonicity* $\delta_V$ at lower noise levels ($\sigma_v$) over expert trajectories compared to using sparse rewards or just using $\hat{V}(,g) = r_W(s,g)$; **Center:** ORS induces *less noisy estimates* of $\hat{V}(s,g)$ over expert trajectories even for long horizons; and **Right:** ORS rewards over 5000 state-action pairs (denoted as dots) for a single fixed goal (denoted as **x**) *smoothly decay in magnitude* with temporal distance from goal. All plots in this figure are computed over **antmaze-giant-navigate**.

***How does ORS influence value learning?*** We analyze how ORS influences the learning dynamics of the induced value function on **antmaze-giant-navigate** *by improving credit assignment*. Following the setup in Sec. 3.2, we analytically compute $\hat{V}(s,g)$ induced by ORS rewards under varying noise levels $\sigma_v$. Figure 5 **(left)** reports how the average $\delta_V$ varies with $\sigma_v$ for sparse rewards, ORS, and a variant that directly uses $\hat{V}(s,g) = r^W(s,g)$. Relative to sparse rewards, ORS exhibits $\delta_V$ that is an order of magnitude smaller at lower noise levels. We plot the approximate value function $\hat{V}(s,g)$ induced by ORS over the same 5 trajectories in Sec. 3.2 in Fig. 5 **(center)**, showing estimates that are much less noisy than sparse-reward $\hat{V}(s,g)$ over long horizons.

Finally, Figure 5 **(right)** plots ORS rewards for a fixed goal over 5000 uniformly sampled state-action pairs, each state displayed in the figure by its x-y position in the maze. ORS reward magnitudes decay smoothly with increasing temporal distance from goal, with dark dots corresponding to states (eg. ends of trajectories) very distant from goal.

***How do different design choices affect ORS?*** We now examine some key design choices behind ORS. First, we compare ORS with a baseline called L2 that computes the uses L2 distance to goal for reward shaping. Table 4 shows that L2 rewards are ineffective and can hurt performance as shown by GCIQL with L2 rewards performing worse than sparse-reward GCIQL.

Next, we ablate over design choices of the ORS reward function and $Q$-function by comparing ORS against a version that uses state-only rewards $r_W(s, g)$ named ORS-s and a variant that uses the reward function itself as the $Q$-function, i.e., $Q(s, a, g) = r_W(s, a, g)$ named ORS-r. We keep the base algorithm the same.

| Dataset | ORS | L2 | GCIQL |
|---|---|---|---|
| antmaze-giant-navigate | $56 \pm 9$ | $3 \pm 2$ | $0 \pm 0$ |
| cube-triple-play | $37 \pm 8$ | $3 \pm 1$ | $7 \pm 3$ |

Table 4: ORS vs L2 rewards

Figure 6: ORS **vs** ORS-s **vs** ORS-r over 5 testtime goals and 8 seeds on **scene-play** (error bars show 95% bootstrapped CI)

In experiments on the manipulation task **scene-play** in Fig. 6, ORS performs best, showing the importance of learning a $Q$-function that is a cumulative sum over $r_W(s, a, g)$, especially when the task involves stitching over trajectories to learn the optimal $Q^*$. We hypothesize that the poor performance of ORS-r is due to noisy estimates of the $Q$ function, as evidenced by our analysis in Fig. 5.

We also ablate the performance of ORS over varying values of the expectile parameter $\kappa$ of GCIQL which is crucially linked to the density of rewards (Kostrikov et al., 2021). The results in Table 5 show higher performance at lower $\kappa$ in locomotion tasks. We hypothesize that this is due to the rich learning signal from dense rewards of ORS. In manipulation tasks, as task complexity increases, the $\kappa$ at which we get best performance increases.

## 6 CONCLUSION

In this paper, we show how generative world models implicitly capture world geometry and introduce Occupancy Reward Shaping, a novel reward-shaping method for offline GCRL that utilizes a

| Dataset | $\kappa$=0.6 | $\kappa$=0.75 | $\kappa$=0.9 |
|---|---|---|---|
| antmaze-giant-navigate | $56 \pm 9$ | $15 \pm 6$ | $1 \pm 1$ |
| puzzle-4x4-play | $70 \pm 5$ | $40 \pm 4$ | $21 \pm 3$ |
| scene-play | $70 \pm 5$ | $80 \pm 4$ | $71 \pm 7$ |
| cube-triple-play | $4 \pm 3$ | $27 \pm 12$ | $37 \pm 8$ |

Table 5: Performance of ORS over varying values of $\kappa$.

generative occupancy model to extract global goal-reaching information into a goal-conditioned reward function. Unlike prior work that relies on graph-building, ORS scales effectively with task complexity and enables learning performative policies over diverse datasets. ORS is simple to implement on top of existing GCRL algorithms and achieves state-of-the-art results over challenging tasks, including real-world Tokamak control.

**Limitations.** Reward shaping methods, by virtue of being sample-based, are less effective in long-horizon tasks with highly combinatorial state spaces and very few ways to achieve task success (e.g. a large multi-step combination lock puzzle). In such regimes, particularly in the offline setting, even ORS rewards may provide limited signal since the occupancy measure has a high Wasserstein distance from goal for almost all states. This challenge could potentially be addressed by learning an occupancy measure over a filtered set of "useful" future states and by augmenting the ORS reward with local rewards that capture short-range state dependencies.

ACKNOWLEDGEMENTS

We would like to thank Grace Liu and Mahsa Bastankhah for reviewing the paper. This material is based upon work supported in part by the U.S. Department of Energy, Office of Fusion Energy Sciences, under Award DE-SC0024544 and on work supported by the National Science Foundation under Award No. 2441665. Any opinions, findings and conclusions or recommendations expressed in this material are those of the author(s) and do not necessarily reflect the views of the National Science Foundation.

**LLM usage:** We used LLMs to aid in LaTeX formatting while writing the paper.

**Reproducibility Statement:** We make our code available on Github. To ensure reliable comparison, each RL algorithm is evaluated over 8 random seeds and 95% bootstrapped CIs are reported

as error bars in tables and plots. Additional information on baselines is provided in Appendix B.2-B.6. We state this information in Sec. 5.1 of the main paper. All proofs are in Appendix A and assumptions are stated in Appendix A.3. We state this in Sec. 4.2 of the main paper.

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

# Appendices

## A    PROOFS

We introduce notation in Sec. A.1. We then provide common setup used for all proofs in Sec. A.2. We introduce the assumptions used in Sec. A.3. Sec. A.4, A.5 and A.6 provide the proofs for Proposition1, Proposition2 and Theorem1, respectively.

### A.1    NOTATION

- **Shortest-path distance**: We use $\text{step}^*(s,g)$ to denote the minimum number of steps to reach $g$ from $s$. We define level sets:
$$\mathcal{S}_k = \{s \in \mathcal{S} : \text{step}^*(s,g) = k\}, \quad k = 0,1,2,\ldots$$
Clearly, $\mathcal{S}_0 = \{g\}$. Under deterministic dynamics, there exists an action $a^*$ at state $s$ such that $p(.\mid s, a^*) \in \mathcal{S}_{k-1}$ whenever $s \in \mathcal{S}_k$, $k \geq 1$.

### A.2    COMMON SETUP FOR PROOFS

The squared Wasserstein-2 distance between two probability measures $\mu$ and $\nu$ is defined as:
$$W_2^2(\mu,\nu) = \inf_{\lambda \in \Lambda(\mu,\nu)} \int \|x-y\|^2 d\lambda(x,y)$$
where $\Lambda(\mu,\nu)$ is the set of all joint measures with marginals $\mu$ and $\nu$.

We consider the special case where one measure is a Dirac delta, $\mu = \delta_g$, located at a single point $g$. In this case, any valid transport plan must move all mass from the point $g$. This removes the need for optimization, as the transport plan is uniquely determined to be the product measure $\Lambda(x,y) = \delta_g(x)\nu(y)$.

Substituting this unique plan into the definition yields:
$$W_2^2(\delta_g,\nu) = \int\int \|y-x\|^2 \delta_g(x)\nu(y)dxdy$$
$$= \int \|y-g\|^2 \nu(y)dy \quad \text{(by the sifting property of } \delta_g(x))$$

The result is the expected squared Euclidean distance from the point $g$ to the distribution $\nu$.

By replacing the general measure $\nu$ with the specific transition probability $d^{\pi_\mathcal{D}}(s^+ \mid s,a)$ and renaming the integration variable, we obtain the final form:
$$W_2^2(\delta_g, d^{\pi_\mathcal{D}}(s^+ \mid s,a)) = \int \|s^+ - g\|^2 d^{\pi_\mathcal{D}}(ds^+ \mid s,a) = M^{\pi_\mathcal{D}}(s,a,g)$$

With this setup, we now define
$$\Phi(s,g) := \|s-g\|^2, \qquad M^{\pi_\mathcal{D}}(s,a,g) := \int \Phi(s^+,g)\, d^{\pi_\mathcal{D}}(ds^+ \mid s,a).$$

Then
$$r^W(s,a,g) = -M^{\pi_\mathcal{D}}(s,a,g).$$

Recall the expanded form of $d^\pi$:
$$d^{\pi_\mathcal{D}}(s^+ \mid s,a) = (1-\gamma)\,p(s' \mid s,a) + \gamma\,d^{\pi_\mathcal{D}}(s^+ \mid s',a'), \quad \forall(s,a,s',a') \in \mathcal{D}$$
Now, substitute into the definition of $M^\pi$ as an integral, and under the assumption of deterministic dynamics:
$$M^{\pi_\mathcal{D}}(s,a,g) = \int \Phi(s^+,g)\, d^{\pi_\mathcal{D}}(ds^+ \mid s,a)$$
$$= \int \Phi(s^+,g)\left[(1-\gamma)\delta_{s'}(s^+ = s') + \gamma d^{\pi_\mathcal{D}}(s^+ \mid s',a')\right]ds^+$$

Using the linearity of the integral, we decompose the above expression into two terms:

$$M^{\pi_{\mathcal{D}}}(s, a, g) = (1 - \gamma)\Phi(s', g) + +\gamma \int \Phi(s^+, g) d^{\pi_{\mathcal{D}}}(s^+ \mid s', a') \, ds^+$$
$$= (1 - \gamma)\Phi(s', g) + \gamma M^{\pi_{\mathcal{D}}}(s', a', g).$$

It is notable that $M^{\pi_{\mathcal{D}}}(s, a, g)$ resembles a squared goal-conditioned Wasserstein-2 distance analogue of the successor representation (Machado et al., 2023).

We also define the state-only version $M^{\pi_{\mathcal{D}}}(s, g) = \mathbb{E}_{a \sim \pi_{\mathcal{D}}(.|s)} M^{\pi_{\mathcal{D}}}(s, a, g)$

### A.3 ASSUMPTIONS

This section lists the assumptions we use to develop our proofs:

**Assumption A.1** (Deterministic dynamics). *We assume that the environment dynamics are deterministic: For any state $s \in \mathcal{S}$ and action $a \in \mathcal{A}$, the state transition probability $p(s' \mid s, a)$ is a point mass distribution, concentrated on the unique successor state $f(s, a)$. That is,*

$$p(\cdot \mid s, a) = \delta_{f(s,a)}(\cdot).$$

**Remark A.1.** *This is a common assumption made for the sake of analytical tractability in foundational reinforcement learning theory. It allows for a clear analysis of the core properties of value functions and policies without the complexities introduced by stochasticity. This assumption serves as a standard basis for theoretical analysis in related work, for instance, in Hartikainen et al. (2020).*

**Assumption A.2** (One-step reachability).

$$\forall k \geq 1, \ \forall s \in S_k, \ \exists a^*(s) \text{ such that } s'|s, a^* \in S_{k-1}.$$

**Remark A.2.** *This assumption is inherent to the definition of a well-posed shortest-path problem. It simply formalizes that the goal is reachable and a shortest path exists from all relevant states.*

**Assumption A.3.** *The potential function $\Phi(s, g)$ is strongly monotonic with respect to the shortest-path distance $step^*(s, g)$. For any two states $s_a, s_b$, if $step^*(s_a, g) \leq step^*(s_b, g)$, then*

$$\Phi(s_a, g) \leq \Phi(s_b, g).$$

*Furthermore, for any $k \geq 1$, if $s_1 \in S_{k-1}$ and $s_2 \in S_k$, there exists a constant $\Delta_\Phi > 0$ such that*

$$\Phi(s_2, g) \geq \Phi(s_1, g) + \Delta_\Phi.$$

**Remark A.3.** *This assumption ensures that the signal provided by the Wasserstein distance is strong and informative, providing a non-trivial cost increase when moving away from the goal.*

**Assumption A.4** (Layer monotonicity of $\pi_{\mathcal{D}}$). *We assume the offline policy $\pi_{\mathcal{D}}$ is consistent with the shortest-path layer structure. For any $k \geq 1$, if $s_1 \in S_{k-1}$ and $s_2 \in S_k$, their successors under $\pi_{\mathcal{D}}$, $s_1' = f(s_1, \pi_{\mathcal{D}}(s_1))$ and $s_2' = f(s_2, \pi_{\mathcal{D}}(s_2))$ must maintain their relative layer ordering. That is,*

$$step^*(s_1', g) \leq step^*(s_2', g).$$

**Remark A.4.** *Assumption A.4 is a condition on the quality of the offline data, which is a standard requirement for obtaining theoretical guarantees in the offline RL setting (Zhan et al., 2022).*

A.4  MONOTONICITY OF GOAL-CONDITIONED REWARD FUNCTION (PROPOSITION 1)

**Proposition.** *Under suitable assumptions on goal reachability, dynamics, and dataset coverage, for all $(s, a, g) \in \mathcal{D}$, the average squared Wasserstein-2 distance*

$$W_2^2\big(\delta_g, d^{\pi_\mathcal{D}}(s^+ \mid s)\big) := \mathbb{E}_{a \sim \pi_\mathcal{D}(\cdot \mid s)}\Big[W_2^2\big(\delta_g, d^{\pi_\mathcal{D}}(s^+ \mid s, a)\big)\Big]$$

*is monotonically increasing with respect to the shortest-path layer index $k$. In particular, for a fixed goal $g$ and any $k \geq 1$, if $s_1 \in \mathcal{S}_{k-1}$ and $s_2 \in \mathcal{S}_k$, then*

$$W_2^2\big(\delta_g, d^{\pi_\mathcal{D}}(s^+ \mid s_1)\big) \leq W_2^2\big(\delta_g, d^{\pi_\mathcal{D}}(s^+ \mid s_2)\big).$$

*Moreover, for any state–goal pair $(s, g)$, the squared Wasserstein-2 distance is minimized by the optimal action:*

$$W_2^2\big(\delta_g, d^{\pi_\mathcal{D}}(s^+ \mid s, a^*)\big) \leq W_2^2\big(\delta_g, d^{\pi_\mathcal{D}}(s^+ \mid s, a)\big), \quad \forall a \in \mathcal{A}, \ a^* \sim \pi^*(\cdot \mid s, g).$$

We prove the proposition in two parts: We first prove that the average $W_2^2$ distance is monotonically decreasing in shortest-path distance to goal and then show that $W_2^2(\delta_g, d^{\pi_\mathcal{D}}(s^+ \mid s, a^*)) \leq W_2^2(\delta_g, d^{\pi_\mathcal{D}}(s^+ \mid s, a))$ for any $(s, a, g)$. Recall that we define the shortest-path distance in terms of the layer index $k$ and show that $W_2^2(\delta_g, d^{\pi_\mathcal{D}}(s^+ \mid s, a)) = M^{\pi_\mathcal{D}}(s, a, g)$ in Sec. A.2. We can now re-write the first part of the proposition as:

**1** (Layer Monotonicity of $M^{\pi_\mathcal{D}}$). *Under Assumptions A.1, A.3 and A.4, the function $M^{\pi_\mathcal{D}}$ is monotonically increasing with respect to the shortest-path layer index $k$. For goal $g$ and any $k \geq 1$, if $s_1 \in S_{k-1}$ and $s_2 \in S_k$, we have:*

$$M^{\pi_\mathcal{D}}(s_1, g) \leq M^{\pi_\mathcal{D}}(s_2, g).$$

It is important to note that the indices $k$ here are such that $k = 0$ refers to the goal state, i.e. index $k - 1$ defines a state closer to goal than index $k$.

*Proof.* Let us define the Bellman operator $T^{\pi_\mathcal{D}}$ which maps a function $M$ to a new function $T^{\pi_\mathcal{D}} M$:

$$(T^{\pi_\mathcal{D}} M)(s, g) = (1 - \gamma)\Phi(f(s, \pi_\mathcal{D}(s)), g) + \gamma M(f(s, \pi_\mathcal{D}(s)), g).$$

We prove this statement by analyzing the properties of the Bellman evaluation operator $T^{\pi_\mathcal{D}}$ associated with the policy $\pi_\mathcal{D}$.

The function $M^{\pi_\mathcal{D}}$ is the unique fixed point of this operator, satisfying $M^{\pi_\mathcal{D}} = T^{\pi_\mathcal{D}}(M^{\pi_\mathcal{D}})$. The existence and uniqueness of this fixed point, and the convergence of value iteration ($M_{i+1} = T^{\pi_\mathcal{D}} M_i$) to it from any starting function $M_0$, is guaranteed by the Banach Fixed-Point Theorem. This is because the operator $T^{\pi_\mathcal{D}}$ is a *contraction mapping* due to the discount factor $\gamma < 1$.

To show this, we take any two functions $M_a, M_b$ and measure the distance between their outputs under the supremum norm $\|M_a - M_b\|_\infty = \max_s |M_a(s) - M_b(s)|$:

$$\|T^{\pi_\mathcal{D}} M_a - T^{\pi_\mathcal{D}} M_b\|_\infty = \max_s \big|(1 - \gamma)\Phi(s', g) + \gamma M_a(s', g) - \big((1 - \gamma)\Phi(s', g) + \gamma M_b(s', g)\big)\big|$$

$$= \max_s \gamma |M_a(s', g) - M_b(s', g)| \quad \text{where } s' = f(s, \pi_\mathcal{D}(s))$$

$$\leq \gamma \max_{z \in \mathcal{S}} |M_a(z, g) - M_b(z, g)|$$

$$= \gamma \|M_a - M_b\|_\infty.$$

Since $\gamma < 1$, the operator is a contraction. Therefore, the sequence converges to the unique fixed point $M^{\pi_\mathcal{D}}$.

Now, we prove that the operator $T^{\pi_\mathcal{D}}$ preserves the property of monotonicity. Let $M$ be an arbitrary function that is monotonically increasing with respect to the layer index $k$. We must show that the resulting function $T^{\pi_\mathcal{D}} M$ is also monotonic.

Let $s_1 \in S_{k-1}$ and $s_2 \in S_k$. Let their successors be $s_1' = f(s_1, \pi_\mathcal{D}(s_1))$ and $s_2' = f(s_2, \pi_\mathcal{D}(s_2))$. We want to show that $(T^{\pi_\mathcal{D}} M)(s_1, g) \leq (T^{\pi_\mathcal{D}} M)(s_2, g)$.

Consider the difference:
$$(T^{\pi_{\mathcal{D}}}M)(s_2, g) - (T^{\pi_{\mathcal{D}}}M)(s_1, g) = (1-\gamma)[\Phi(s_2', g) - \Phi(s_1', g)] + \gamma[M(s_2', g) - M(s_1', g)].$$

By Assumption A.4, we have $\text{step}^*(s_1', g) \leq \text{step}^*(s_2', g)$. Now we analyze the two terms in the difference:

- By Assumption A.3, the condition $\text{step}^*(s_1', g) \leq \text{step}^*(s_2', g)$ directly implies that $[\Phi(s_1', g) \leq \Phi(s_2', g)]$. The first term $[\Phi(s_2', g) - \Phi(s_1', g)]$ is non-negative.

- By our premise for this step, the function $M$ is monotonic. Since $d^*(s_1', g) \leq d^*(s_2', g)$, it follows that $M(s_1', g) \leq M(s_2', g)$. Thus, the second term $[M(s_2', g) - M(s_1', g)]$ is also non-negative.

Since both terms are non-negative, their sum is non-negative, and thus $(T^{\pi_{\mathcal{D}}}M)(s_1, g) \leq (T^{\pi_{\mathcal{D}}}M)(s_2, g)$. This proves that the operator $T^{\pi_{\mathcal{D}}}$ preserves monotonicity.

The value iteration process starts with an initial value function $M_0(s) = 0$ for all $s$. The zero function is trivially monotonic. Since the operator $T^{\pi_{\mathcal{D}}}$ preserves monotonicity, the entire sequence of value functions $\{M_0, M_1, M_2, \dots\}$ generated by $M_{i+1} = T^{\pi_{\mathcal{D}}}M_i$ consists of monotonic functions. The final value function $M^{\pi_{\mathcal{D}}}$ is the limit of this sequence, and the limit of a sequence of monotonic functions is also monotonic.

Therefore, $M^{\pi_{\mathcal{D}}}$ is monotonically increasing with respect to the layer index $k$.

It follows that average $W_2^2$ distance $\mathbb{E}_{a \sim \pi_{\mathcal{D}}(\cdot|s)} W_2^2(\delta_g, d^{\pi_{\mathcal{D}}}(s^+ \mid s, a))$ is monotonically decreasing in shortest-path distance towards goal $g$.

$\square$

We now prove the second part. Similar to part 1, we can re-write in terms of $M^{\pi_{\mathcal{D}}}$ as follows:

**2** (Optimal actions under $M^{\pi_{\mathcal{D}}}(s, a, g)$)**.** *Under Assumptions A.1-A.4 and the layer monotonicity of $M^{\pi_{\mathcal{D}}}$, for any state $s \in S_k$ with $k \geq 1$, any optimal, shortest-path-inducing, action $a^* \sim \pi^*(.|s, g)$, and any non-optimal action $a_{bad}$, the function $M^{\pi_{\mathcal{D}}}$ is strictly smaller for the shortest-path action. That is,*
$$M^{\pi_{\mathcal{D}}}(s, a^*, g) < M^{\pi_{\mathcal{D}}}(s, a_{bad}, g).$$

*Proof.* Let $s^* = f(s, a^*)$ and $s_{bad} = f(s, a_{bad})$. By definition, $s^* \in S_{k-1}$ and $s_{bad} \in S_j$ for some $j \geq k$. We analyze the difference $\Delta M = M^{\pi_{\mathcal{D}}}(s, a_{bad}, g) - M^{\pi_{\mathcal{D}}}(s, a^*, g)$:

$$\Delta M = \Big[(1-\gamma)\Phi(s_{bad}, f) + \gamma M^{\pi_{\mathcal{D}}}(s_{bad}, g)\Big] - \Big[(1-\gamma)\Phi(s^*, g) + \gamma M^{\pi_{\mathcal{D}}}(s^*, g)\Big]$$
$$= \underbrace{(1-\gamma)[\Phi(s_{bad}, g) - \Phi(s^*, g)]}_{\text{Term 1}} + \underbrace{\gamma[M^{\pi_{\mathcal{D}}}(s_{bad}, g) - M^{\pi_{\mathcal{D}}}(s^*, g)]}_{\text{Term 2}}$$

For Term 1, since $s^* \in S_{k-1}$ and $s_{bad} \in S_j$ with $j \geq k$, by Assumption A.3, we have $\Phi(s_{bad}, g) \geq \Phi(s^*, g) + \Delta_\Phi$. Thus, Term 1 is strictly positive and bounded below by $(1-\gamma)\Delta_\Phi > 0$.

For Term 2, since $s^* \in S_{k-1}$ and $s_{bad} \in S_j$ with $j \geq k$, by Lemma 1, we have $M^{\pi_{\mathcal{D}}}(s^*, g) \leq M^{\pi_{\mathcal{D}}}(s_{bad}, g)$. Thus, Term 2 is non-negative.

The sum of a strictly positive term and a non-negative term is strictly positive. Therefore, $\Delta M > 0$.

It follows that $r^W(s, a^*, g) > r^W(s, a_{bad}, g)$ or equivalently, $r^W(s, a^*, g) \geq r^W(s, a, g)$, where $a$ can be any action in $\mathcal{D}$ at state $s$, which concludes the proof. $\square$

## A.5 Estimating squared Wasserstein-2 distance (Proposition 2)

**Proposition.** *The squared Wasserstein-2 distance between $\delta_g$ and $d^{\pi_{\mathcal{D}}}(s^+ \mid s, a)$ is upper bounded, up to a multiplicative constant $C > 0$, by the mean squared error of the associated velocity field corresponding to $d^{\pi_{\mathcal{D}}}(s^+ \mid s, a)$ and the target velocity corresponding to $\delta_g$:*

$$W_2^2\big(\delta_g, d^{\pi_{\mathcal{D}}}(s^+ \mid s, a)\big) \leq C \mathbb{E}_{\substack{x_1 \sim \delta_g \\ x_0 \sim \mathcal{N}(0, I_d) \\ t \sim \text{Unif}([0,1])}} \big\| v(t, s, a, x_t) - (x_1 - x_0) \big\|_2^2, \tag{6}$$

*where $x_t = tx_1 + (1-t)x_0$.*

We note that $x_1 - x_0$ gives us the velocity at $(x_0, x_1)$, where $x_1 = g$. We start from the definition of squared Wasserstein-2 distance in Sec. A.2. In our case, $\mu$ corresponds to $\delta_g$ and $\nu$ corresponds to $d_\theta^{\pi_\mathcal{D}}(s^+ \mid s, a)$. As $\mu = \delta_g$, a point mass centered at $g$, the transport plan is uniquely determined to be the product measure $\lambda(x, s^+) = \delta_g(x)d_\theta^{\pi_\mathcal{D}}(s^+ \mid s, a)$ (Sec. A.2). For a given $(s, a, g)$, the inequality follows from Lemma 1 of Lv et al. (2025) by Gronwall's inequality:

$$
\begin{aligned}
\inf_{\lambda \in \Lambda(\delta_g, d_\theta^{\pi_\mathcal{D}})} \int \|x - s^+\|_2^2 \, d\lambda(x, s^+) &= W_2^2\big(\delta_g, d_\theta^{\pi_\mathcal{D}}(s^+ \mid s, a)\big). \\
&\leq C \, \mathbb{E}_{\substack{g = x_1 \sim \delta_g, \\ x_0 \sim \mathcal{N}(0, I_d), \\ t \sim \mathrm{Unif}([0,1])}} \left[ \big\|v_{d^{\pi_\mathcal{D}}}(t, s, a, x_t) - (x_1 - x_0)\big\|_2^2 \right]
\end{aligned}
\tag{7}
$$

### A.6 OPTIMALITY OF ORS POLICY (PROOF OF THEOREM 1)

**Theorem.** *Under suitable assumptions on goal reachability, dynamics, and dataset coverage, the greedy goal-conditioned policy:*

$$
\pi^{\mathrm{greedy}}(a \mid s, g) = \arg\max_{a \in \mathcal{A}} Q^*(s, a, g),
$$

*where $Q^*(s, a, g)$ denotes the optimal action-value function induced by the reward $r^W(s, a, g)$, coincides with the optimal shortest-path policy $\pi^*(a \mid s, g)$.*

The optimal action-value function, $Q^*(s, a, g)$, for the reward $r^W(s, a, g)$ is the unique fixed point of the Bellman optimality operator:

$$
Q^*(s, a, g) = r^W(s, a, g) + \gamma \max_{a'} Q^*(f(s, a), a', g).
$$

The corresponding optimal state-value function is $V^*(s, g) = \max_{a'} Q^*(s, a', g)$. $\pi^{greedy}$ is the policy that greedily maximizes $Q^*(s, a, g)$:

$$
\pi^{greedy}(a|s, g) := \arg\max_a Q^*(s, a, g) = \arg\min_a \{M^{\pi_\mathcal{D}}(s, a, g) - \gamma V^*(f(s, a), g)\}.
$$

**Lemma A.1** (Layer-wise Monotonicity of the Optimal Value Function $V^*$). *Under Assumptions A.1-A.4 and layer monotonicity of $M^{\pi_\mathcal{D}}$ (Sec. A.4), the value function of the optimal shortest-path policy, $V^*(s, g)$, is strictly monotonically non-decreasing with the shortest-path layer index (i.e., the value is strictly higher for states closer to the goal). For any $k \geq 1$, if $s_1 \in S_{k-1}$ and $s_2 \in S_k$, we have:*

$$
V^*(s_1, g) - V^*(s_2, g) \geq (1 - \gamma^{k-1})\Delta_\Phi > 0.
$$

*Consequently, it holds that $V^*(s_2, g) < V^*(s_1, g)$.*

*Proof.* Let $\pi^*$ denote the optimal shortest-path policy. For the sake of clarity, we write that for any state $s$ and goal $g$, let $a^*(s) = \pi^*(a^*|s, g)$. We first establish the difference in the one-step cost for taking a shortest-path action from two adjacent layers.

Let $s_a \in S_{j-1}$ and $s_b \in S_j$ for any $j \geq 1$. Let their successors under $\pi^*$ be $s_a' = f(s_a, a^*(s_a)) \in S_{j-2}$ and $s_b' = f(s_b, a^*(s_b)) \in S_{j-1}$. Consider the difference in their one-step costs:

$$
\begin{aligned}
\Delta M_{step} &= M^{\pi_\mathcal{D}}(s_b, a^*(s_b), g) - M^{\pi_\mathcal{D}}(s_a, a^*(s_a), g) \\
&= \left[(1-\gamma)\Phi(s_b', g) + \gamma M^{\pi_\mathcal{D}}(s_b', g)\right] - \left[(1-\gamma)\Phi(s_a', g) + \gamma M^{\pi_\mathcal{D}}(s_a', g)\right] \\
&= (1-\gamma)[\Phi(s_b', g) - \Phi(s_a', g)] + \gamma[M^{\pi_\mathcal{D}}(s_b', g) - M^{\pi_\mathcal{D}}(s_a', g)].
\end{aligned}
$$

By Assumption 3 (Strong Monotonicity of $\Phi$), since $s_a' \in S_{j-2}$ and $s_b' \in S_{j-1}$, we have $[\Phi(s_b', g) - \Phi(s_a', g)] \geq \Delta_\Phi$. By Lemma 1 (Layer Monotonicity of $M^{\pi_\mathcal{D}}$), since $s_a' \in S_{j-2}$ and $s_b' \in S_{j-1}$, we have $[M^{\pi_\mathcal{D}}(s_b', g) - M^{\pi_\mathcal{D}}(s_a', g)] \geq 0$. Therefore, we have a lower bound on the single-step cost difference:

$$
M^{\pi_\mathcal{D}}(s_b, a^*(s_b), g) - M^{\pi_\mathcal{D}}(s_a, a^*(s_a), g) \geq (1-\gamma)\Delta_\Phi.
$$

Now, let's consider the two shortest-path trajectories starting from $s_1 \in S_{k-1}$ and $s_2 \in S_k$. Let $\tau_1^* = (s_{1,0}, s_{1,1}, \ldots, s_{1,k-1} = g)$ be the trajectory starting from $s_{1,0} = s_1$. Let $\tau_2^* = (s_{2,0}, s_{2,1}, \ldots, s_{2,k} = g)$ be the trajectory starting from $s_{2,0} = s_2$. By definition of the shortest-path policy, we have $s_{1,t} \in S_{k-1-t}$ and $s_{2,t} \in S_{k-t}$ for $t \in [0, k-1]$.

The value functions are the sum of discounted negative costs:

$$V^*(s_1, g) = -\sum_{t=0}^{k-2} \gamma^t M^{\pi_\mathcal{D}}(s_{1,t}, a^*(s_{1,t}), g)$$

$$V^*(s_2, g) = -\sum_{t=0}^{k-1} \gamma^t M^{\pi_\mathcal{D}}(s_{2,t}, a^*(s_{2,t}), g)$$

Let's analyze the difference $V^*(s_1, g) - V^*(s_2, g)$:

$$V^*(s_1, g) - V^*(s_2, g) = \sum_{t=0}^{k-1} \gamma^t M^{\pi_\mathcal{D}}(s_{2,t}, a^*(s_{2,t}), g) - \sum_{t=0}^{k-2} \gamma^t M^{\pi_\mathcal{D}}(s_{1,t}, a^*(s_{1,t}), g)$$

$$= \gamma^{k-1} M^{\pi_\mathcal{D}}(s_{2,k-1}, a^*(s_{2,k-1}), g) + \sum_{t=0}^{k-2} \gamma^t \Big[ M^{\pi_\mathcal{D}}(s_{2,t}, a^*(s_{2,t}), g) - M^{\pi_\mathcal{D}}(s_{1,t}, a^*(s_{1,t}), g) \Big].$$

For each term in the summation (for $t \in [0, k-2]$), we have $s_{1,t} \in S_{k-1-t}$ and $s_{2,t} \in S_{k-t}$. Applying our single-step cost difference result, we get:

$$M^{\pi_\mathcal{D}}(s_{2,t}, a^*(s_{2,t}), g) - M^{\pi_\mathcal{D}}(s_{1,t}, a^*(s_{1,t}), g) \geq (1-\gamma)\Delta_\Phi.$$

The last term for the $\tau_2^*$ trajectory is $M^{\pi_\mathcal{D}}(s_{2,k-1}, a^*(s_{2,k-1}), g)$. Since $s_{2,k-1} \in S_1$, its successor is $g \in S_0$. We know $M^{\pi_\mathcal{D}} \geq 0$. So we can lower bound the entire difference:

$$V_W^{\pi^*}(s_1, g) - V_W^{\pi^*}(s_2, g) \geq \gamma^{k-1} \cdot 0 + \sum_{t=0}^{k-2} \gamma^t (1-\gamma)\Delta_\Phi$$

$$= (1-\gamma)\Delta_\Phi \sum_{t=0}^{k-2} \gamma^t$$

$$= (1-\gamma)\Delta_\Phi \frac{1-\gamma^{k-1}}{1-\gamma}$$

$$= (1-\gamma^{k-1})\Delta_\Phi.$$

Since $k \geq 1$ and $\gamma \in (0, 1)$, the term $(1 - \gamma^{k-1})$ is strictly positive. As $\Delta_\Phi > 0$, the entire lower bound is strictly positive. This completes the proof. $\qquad\square$

**Theorem A.1** (Equivalence of the Q-learning Optimum and the Shortest Path). *Under Assumptions A.1-A.4 and noting that $r^W(s, a^*, g) > r^W(s, a_{bad}, g)$ (Sec A.4), the greedy policy with respect to $Q^*$ is the optimal shortest-path policy $\pi^*$.*

*Proof.* We must show that for any state $(s, g)$, the shortest-path action $a^* = \pi^*(.|s, g)$ is the unique maximizer of the action-value function $Q^*(s, a, g)$ and is therefore, the same action chosen by $\pi^{greedy}(. \mid s, g)$. That is, for any non-shortest-path action $a_{bad} \neq a^*$:

$$Q^*(s, a^*, g) > Q^*(s, a_{bad}, g).$$

Let's analyze the difference $\Delta Q = Q^*(s, a^*, g) - Q^*(s, a_{bad}, g)$. Let the successor states be $s^* = f(s, a^*)$ and $s_{bad} = f(s, a_{bad})$.

$$\Delta Q = \Big[ r^W(s, a^*, g) + \gamma V^*(s^*, g) \Big] - \Big[ r^W(s, a_{bad}, g) + \gamma V^*(s_{bad}, g) \Big]$$

$$= \Big[ -M^{\pi_\mathcal{D}}(s, a^*, g) + \gamma V^*(s^*, g) \Big] - \Big[ -M^{\pi_\mathcal{D}}(s, a_{bad}, g) + \gamma V^*(s_{bad}, g) \Big]$$

$$= \underbrace{\Big[ M^{\pi_\mathcal{D}}(s, a_{bad}, g) - M^{\pi_\mathcal{D}}(s, a^*, g) \Big]}_{\text{Term 1}} + \underbrace{\gamma \Big[ V^*(s^*, g) - V^*(s_{bad}, g) \Big]}_{\text{Term 2}}$$

We will now analyze the two terms of this expression.

The first term, $M^{\pi_\mathcal{D}}(s, a_{bad}, g) - M^{\pi_\mathcal{D}}(s, a^*, g)$, by part 2 of the proof in Sec. A.4, this term is guaranteed to be positive.

We now analyze the second term, $\gamma[V^*(s^*, g) - V^*(s_{\text{bad}}, g)]$. By definition, the successor state $s^*$ lies in a layer closer to the goal than $s_{\text{bad}}$ (i.e., $d^*(s^*, g) < d^*(s_{\text{bad}}, g)$). It therefore follows directly from Lemma A.1 that

$$V^*(s_{\text{bad}}, g) \ < \ V^*(s^*, g).$$

Thus, the term $\left[V^*(s^*, g) - V^*(s_{\text{bad}}, g)\right]$ is strictly positive.

The difference in Q-values, $\Delta Q$, is the sum of a strictly positive term and a strictly positive term (Term 2). The sum is therefore strictly positive.

$$\Delta Q > 0 \quad i.e \quad Q^*(s, a^*, g) > Q^*(s, a_{bad}, g)$$

This shows that for any state $s$, the shortest-path action $a^* \sim \pi^*(a|s, g)$ is the unique greedy action with respect to $Q^*$. Therefore, the greedy policy with respect to $Q^*$ must be the optimal shortest-path policy $\pi^*$. □

# B ADDITIONAL INFORMATION

## B.1 OGBENCH TASKS

- **Antmaze datasets:** Antmaze involves controlling a quadruped Ant robot with 8 degrees of freedom (DoF) to reach a given goal location in the maze. The agent must learn both high-level navigation and low-level locomotion skills purely from offline data. We perform experiments on 3 types of challenging mazes: **antmaze-large-navigate**, **antmaze-giant-navigate** and **antmaze-large-explore**.

  **antmaze-giant** is twice the size of the large and same size as **antmaze-ultra** in (Jiang et al., 2022) designed to substantially challenge long-horizon planning capabilities. Both **antmaze-large-navigate** and **antmaze-giant-navigate** are collected with noisy expert SAC policies that repeatedly move towards randomly sampled goals (Park et al., 2024a). **antmaze-large-explore** features extremely low quality yet high coverage data consisting of random exploratory trajectories.

- **Cube datasets:** The cube environments involve manipulation of cube blocks using a robot arm. We perform experiments on 2 cube-play datasets which are collected using a scripted policy that repeatedly picks a cube and places it in other random locations or on another cube: **cube-double-play** (2 cubes) and **cube-triple-play** (3 cubes). During evaluation, the agent is required to perform cube moving, stacking, swapping or permutations according to the provided goal configuration. This requires learning generalizable pick-and-place operations with multiple objects and stitching across unstructured data to achieve this. The task horizon also increases with the number of cubes. We also perform experiments on the **cube-triple-noisy** dataset which involves highly sub-optimal data with noisy transitions.

- **Puzzle datasets:** These environments specifically test out the combinatorial generalization abilities of the agent (upto $2^{24}$ puzzle configurations and a much larger state space due to robotic manipulation on the puzzle) under very long horizons. They require solving the Lights Out puzzle (Park et al., 2024a), which consists of a 2D grid with buttons in cells, where pressing buttons toggles the color the that button and buttons around it. The agent has to achieve an appropriate goal configuration of colors by pressing buttons by controlling a robot arm. We evaluate algorithms on play datasets of **puzzle-4x4**, **puzzle-4x5** and **puzzle-4x6**; and noisy datasets of **puzzle-4x4** and **puzzle-4x6**. These are particularly challenging tasks (Park et al., 2024a).

- **Scene datasets:** The scene tasks challenge sequential, long-horizon planning. They involve manipulating diverse objects: a cube, a window, a drawer and two locks. **scene-play** features a dataset collected using a scripted policy that randomly interacts with the objects. At test-time, the agent has to arrange objects into a given goal configuration. These tasks can involve up to 8 sub-tasks e.g. unlock a drawer, then open it, then put a cube in it and close it again, etc. The agent must therefore be able to stitch together multiple skills to

achieve success. **scene-noisy** involves the same tasks but the dataset is of substantially lower quality.

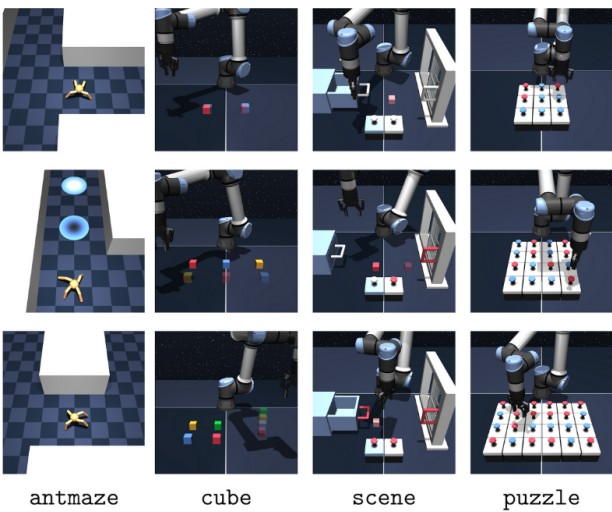

antmaze          cube          scene          puzzle

Figure 7: OGBench tasks.

## B.2 TOKAMAK TASKS

A tokamak (Figure 8) is a scientific device that uses electro-magnetic fields to confine extremely hot ionized gas called plasma in a toroidal chamber, with the aim of achieving controlled nuclear fusion It is considered the most promising experimental design for achieving commercial nuclear fusion at scale, opening up avenues for nearly unlimited clean energy.

We use a dataset of raw sensor and actuator data collected from the DIII-D tokamak located in San Diego, CA, USA. Each trajectory corresponds to one plasma discharge, with a control frequency of 25 ms. A control task corresponds to tracking certain state variables of the plasma through the duration of a discharge. While we were not able to get experiment time on the reactor on short notice, we evaluated each RL algorithm using rollouts from a thoroughly tested and accurately learned model of plasma dynamics (Char et al., 2023; 2024) as a proxy. Each RL algorithm was evaluated based on how closely it tracks a given goal state of the plasma. The reward at each time-step during evaluation corresponds to the L2 distance between the goal variables to track and the actual quantities achieved through control.

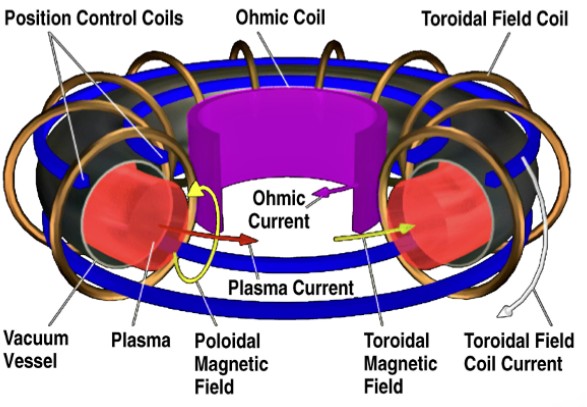

Figure 8: Tokamak fusion reactor from (Walker et al., 2020).

We evaluated each algorithm over 10 goals. The offline dataset size was roughly 50k samples after data cleaning and processing. The state and action spaces are outlined in the following table:

| Category | Variables |
|---|---|
| State Space | **Scalar States:** $\beta_N$, Internal Inductance, Line Averaged Density, Loop Voltage, Stored Energy
**Profile States:** Electron Density, Electron Temperature, Pressure, Safety Factor, Ion Temperature, Ion Rotation |
| Action Space | Power Injected, Torque Injected, Total Deuterium Gas Injection, Total ECH Power |

Please refer to Abbate et al. (2021); Char et al. (2023; 2024) for detailed explanations of the specific variables and actuators.

### B.3 ADDITIONAL INFORMATION ON BASELINES

- **For baselines provided in the OGBench (Park et al., 2024a) repository, we use the best hyperparameters provided in the paper. These include: GCBC, GCIVL, GCIQL, QRL, CRL and HIQL**.

- **GCIQL**: GCIQL is a goal-conditioned variant of Implicit Q Learning (Kostrikov et al., 2021) that fits the optimal $Q^*$ using expectile regression (Newey & Powell, 1987). GCIQL learns $Q$ and $V$ as follows:

$$\mathcal{L}^V_{\text{GCIQL}}(V) = \mathbb{E}_{(s,a)\sim p^\mathcal{D}(s,a),\, g\sim p^\mathcal{D}_{\text{mixed}}(g|s)} \left[\ell^2_\kappa\left(\bar{Q}(s,a,g) - V(s,g)\right)\right],$$

$$\mathcal{L}^Q_{\text{GCIQL}}(Q) = \mathbb{E}_{(s,a,s')\sim p^\mathcal{D}(s,a,s'),\, g\sim p^\mathcal{D}_{\text{mixed}}(g|s)} \left[\left(r^W(s,a,g) + \gamma V(s',g) - Q(s,a,g)\right)^2\right]$$

Here, $\bar{Q}$ is the target Q function (Mnih et al., 2015), $\kappa$ is the expectile and $r^W(s,a,g)$ is the ORS reward. For policy extraction, we use deterministic policy gradient with a behavioral regularization (**DDPG + BC**) (Fujimoto & Gu, 2021):

$$J_{\text{DDPG+BC}}(\pi) = \mathbb{E}_{(s,a)\sim p^\mathcal{D}(s,a),\, g\sim p^\mathcal{D}_{\text{mixed}}(g|s)} \left[Q(s,\pi^\mu(s,g),g) + \alpha \log \pi(a \mid s,g)\right]$$

where $\pi^\mu(s,g) = \mathbb{E}_{a\sim\pi(a|s,g)}[a]$. We use double-Q learning and normalize the Q values before policy extraction.

- **Go-Fresh** (Mezghani et al., 2023): Go-Fresh involves 3 stages: Training the R-Net, a neural network that learns local temporal distances between states, using R-Net and R-Net embeddings to create a semi-parametric graph of the offline dataset and then training a policy. While training a policy, every 1000 iterations of training, Go-Fresh creates a new dataset where each tuple corresponds to a randomly sampled $(s,g)$ pair from the dataset.

  We noticed in initial experiments that even when Go-Fresh is trained using the same GCRL algorithm as ORS (GCIQL), overall success-rate was close to 0 across tasks. To ensure a fair and unbiased comparison, we modified the Go-Fresh codebase to have the same goal sampling strategy from OGBench (detailed in Appendix D of (Park et al., 2024a) and in the last item of this sub-section) used for all other baselines. This substantially improved performance and is required to produce non-negligible performance, as reported in Tables 1 and 2.

  Furthermore, noticing that Go-Fresh is tested on relatively simpler tasks compared to ours in their paper (Mezghani et al., 2023), we modify the following hyperparameters to account for the increased task difficulty, in addition to using best values for other hyperparameters:

  We used a memory capacity of 5000 on all tasks which we found to be sufficient (the graph building stage consistently produced a graph with less than 5000 nodes). We increased the R-Net size to an MLP of 64 units (we did not see an improvement in classification accuracy for large sizes) and used a local distance threshold $\tau = 10$ on all tasks. We also see that weighing the local reward by a factor of 5 before summing it with the global reward gave better performance.

- **Goal sampling for all algorithms**: The same goal-sampling method was used for all baselines in this paper as well as ORS and all variations. Goals were sampled in three different ways according to probabilities from a categorical distribution defined as follows:

– $p_{\mathrm{cur}}^{\mathcal{D}}(g|s)$: Dirac delta distribution centered at the current state $s$.

– $p_{\mathrm{traj}}^{\mathcal{D}}(g|s)$: goal is sampled from future states of $s$ from the same trajectory with uniform sampling.

– $p_{\mathrm{rand}}^{\mathcal{D}}(g|s)$: goal is uniformly sampled from $\mathcal{D}$.

We provide information on the specific values used in Sec. B.6 and B.7 and Table 6.

- **n-step returns:** We implement two ways of doing n-step returns in the critic with sparse rewards (where $r(s, g) = -\mathbf{1}(s \neq g)$ and $t$ is the MDP timestep):

  1. **n-step GCIQL**: $Q(s_t, a_t, g) = \sum_{i=t}^{t+n-1} r(s_{t+i}, g) + \gamma^n Q(s_{t+n}, \pi(a_{t+n}|s_{t+n}, g), g)$

  2. **GCIQL-OTA** (based on the core idea in (Ahn et al., 2025)):
     $Q(s_t, a_t, g) = -\mathbf{1}(s_{t+n} \neq g) + \gamma Q(s_{t+n}, \pi(a_{t+n}|s_{t+n}, g), g)$

As evident, we use GCIQL as the base algorithm for n-step return algorithms. We provide the specific hyperparameters used for each task in Table 6. Please refer to Sec. B.7 for information on all other common hyperparameters.

Table 6: Specific hyperparameters for best performance with **n-step GCIQL** and **GCIQL-OTA**

| Dataset | $\alpha$ (BC-coefficient) | $\kappa$ (Expectile) | n | $\gamma$ | Actor $(p_{\mathrm{cur}}^{\mathcal{D}}, p_{\mathrm{traj}}^{\mathcal{D}}, p_{\mathrm{rand}}^{\mathcal{D}})$ |
|---|---|---|---|---|---|
| puzzle-4x6-play | 1.0 | 0.9 | 25 | 0.995 | (0, 1, 0) |
| cube-triple-play | 3.0 | 0.9 | 50 | 0.995 | (0, 1, 0) |
| antmaze-large-navigate | 0.3 | 0.9 | 50 | 0.995 | (0, 1, 0) |
| antmaze-giant-navigate | 0.3 | 0.9 | 50 | 0.995 | (0, 1, 0) |
| puzzle-4x6-noisy | 0.03 | 0.9 | 25 | 0.995 | (0, 1, 0) |
| cube-triple-noisy | 0.03 | 0.9 | 50 | 0.995 | (0, 1, 0) |
| antmaze-large-explore | 0.01 | 0.9 | 50 | 0.995 | (0, 0, 1) |

## B.4 TRAJECTORY VISUALIZATIONS: ANALYSED TRAJECTORIES

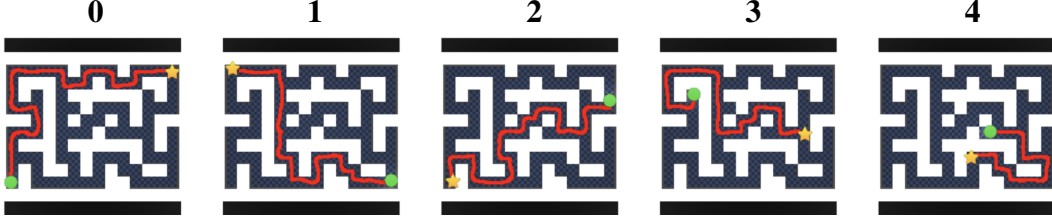

Figure 9: Expert trajectories of varying lengths on **antmaze-giant-navigate** used for analysis in Sec. 3.2 and Sec. 5.1. The green dot represents the starting position and the star represents the goal.

## B.5 TRAINING ITERATIONS

Table 7: Training iterations per task. All algorithms are trained for the same number of iterations.

| Dataset | Training Iterations |
|---|---|
| cube-double-play | 1M |
| puzzle-4x4-play | 1M |
| scene-play | 1M |
| puzzle-4x5-play | 1M |
| puzzle-4x6-play | 1M |
| cube-triple-play | 2M |
| antmaze-large-navigate | 3M |
| antmaze-giant-navigate | 6M |
| puzzle-4x4-noisy | 1M |
| scene-noisy | 1M |
| puzzle-4x6-noisy | 1M |
| cube-triple-noisy | 1M |
| antmaze-large-explore | 6M |

## B.6 OCCUPANCY AND REWARD MODEL HYPERPARAMETERS

Table 8: Occupancy and reward-model hyperparameters.

| Hyperparameter | Value |
|---|---|
| Gradient steps | 2M |
| Optimizer | Adam (Kingma & Ba, 2014) |
| Learning rate | 0.0003 |
| Batch size | 256 |
| Architecture | MLP |
| MLP size | $\{512, 512, 512, 512\}$ |
| Nonlinearity | GELU (Hendrycks & Gimpel, 2016) |
| Layer normalization | True |
| Target net update rate (occupancy model) | 0.005 |
| Discount factor $\gamma$ (occupancy model) | 0.99 |
| Flow steps | 22 (antmaze-giant-navigate), 55 (antmaze-large-navigate), 45 (other tasks) |
| Reward ($p_{\text{cur}}^{\mathcal{D}}, p_{\text{traj}}^{\mathcal{D}}, p_{\text{rand}}^{\mathcal{D}}$) ratio for goal sampling | (0.2, 0.5, 0.3) |

We added a warm-start stage without bootstrapping to the occupancy model by training it to predict future states $s^+$ distributed according to a geometric distribution over trajectory timesteps $t_\tau \sim \text{Geom}(1 - \gamma)$ starting the current state $s$:

$$\mathcal{L}_{pretrain}(\theta) = \mathbb{E}_{\substack{s^{t_\tau} = x_1 \sim \mathcal{D}, t_\tau \sim \text{Geom}(1-\gamma), \\ x_0 \sim \mathcal{N}(0, I_d), t \sim \text{Unif}([0,1])}} \left[ \| v_\theta(t, s, a, x_t) - (x_1 - x_0) \|_2^2 \right] \quad (8)$$

for the first 1M epochs and then training it using the bootstrapping loss in Sec. 4.1 for next next 1M epochs, however we did not see any consistent improvements from doing this.

## B.7 POLICY TRAINING HYPERPARAMETERS

We first list hyperparameters common to all algorithms in Table 9:

Table 9: Common hyperparameters for policy training.

| Hyperparameter | Value |
|---|---|
| Optimizer | Adam (Kingma & Ba, 2014) |
| Learning rate | 0.0003 |
| Batch size | 1024 |
| Architecture | MLP |
| MLP size (Actor and Critic) | {512, 512, 512} |
| Nonlinearity | GELU (Hendrycks & Gimpel, 2016) |
| Layer normalization | True |
| Target critic update rate | 0.005 |
| Critic $(p_{\text{cur}}^{\mathcal{D}}, p_{\text{traj}}^{\mathcal{D}}, p_{\text{rand}}^{\mathcal{D}})$ ratio for goal sampling | (0.2, 0.5, 0.3) |

We then provide specific hyperparameters used for ORS and Go-Fresh (which use GCIQL as the base algorithm to train policies) in Table 10:

Table 10: Specific hyperparameters for policy training: ORS and Go-Fresh

| Dataset | $\alpha$ (BC-coefficient) | $\kappa$ (Expectile) | $\gamma$ | Actor $(p_{\text{cur}}^{\mathcal{D}}, p_{\text{traj}}^{\mathcal{D}}, p_{\text{rand}}^{\mathcal{D}})$ | reward scaling for ORS (linear scaling $x : (r/x)$) |
|---|---|---|---|---|---|
| puzzle-4x4-play | 0.6 | 0.6 | 0.99 | (0, 1, 0) | 0.75 |
| puzzle-4x5-play | 0.6 | 0.6 | 0.99 | (0, 0.5, 0.5) | 0.75 |
| puzzle-4x6-play | 0.6 | 0.9 | 0.99 | (0, 0.5, 0.5) | 0.25 |
| cube-double-play | 0.3 | 0.75 | 0.99 | (0, 0.5, 0.5) | 0.25 |
| cube-triple-play | 0.3 | 0.9 | 0.99 | (0, 0.5, 0.5) | 0.25 |
| scene-play | 0.3 | 0.75 | 0.99 | (0, 0.5, 0.5) | 0.75 |
| antmaze-large-navigate | 0.15 | 0.6 | 0.995 | (0, 1, 0) | 2.0 |
| antmaze-giant-navigate | 0.1 | 0.6 | 0.995 | (0, 1, 0) | 2.0 |
| puzzle-4x4-noisy | 0.05 | 0.6 | 0.99 | (0, 1, 0) | 0.25 |
| puzzle-4x6-noisy | 0.05 | 0.75 | 0.99 | (0, 1, 0) | 0.25 |
| cube-triple-noisy | 0.03 | 0.75 | 0.99 | (0, 0.5, 0.5) | 1.0 |
| scene-noisy | 0.1 | 0.75 | 0.99 | (0, 1, 0) | 0.25 |
| antmaze-large-explore | 0.015 | 0.9 | 0.99 | (0, 0.5, 0.5) | 3.0 |

## B.8 WALL-CLOCK TIME

The average wall-clock time per iteration is as follows:

| Method | Component | Time per iteration (ms) |
|---|---|---|
| ORS | Occupancy model | 14 |
| | Reward function | 8 |
| | Policy training | 6.5 |
| GO-FRESH | R-NET training | 9.5 |
| | Policy training | 7.2 |

GO-FRESH also has a graph computation time that ranges between 1.5-10 mins depending on the task and dataset. For default GCIQL, policy training takes 7.2 ms per iteration. All times were reported on a single NVIDIA RTX A6000 GPU. We typically see that reward function and occupancy model losses converge in around 500k iterations. R-Net training typically runs for 500k iterations.

While ORS does lead to a slightly higher computational overhead, it also leads to a 2.3 $\times$ increase in performance over GCIQL, its base algorithm, and exhibits stronger performance than GO-FRESH on a variety of tasks. GO-FRESH requires training a distance classifier, training a graph memory and building a graph, each of which involves hyperparameter tuning for the size of the R-Net, number of edges in the graph, number of negative pairs for training the R-Net, distance threshold for the R-Net and careful weighing of local and global rewards during policy training. In contrast, ORS is simple to train, does not require storing a large graph in memory and is easy to tune. We agree that increasing the efficiency of ORS is an important direction for future work.

