# OpenReview forum: "Occupancy Reward Shaping: Improving Credit Assignment for Offline Goal-Conditioned Reinforcement Learning"
_ICLR.cc/2026/Conference — ICLR 2026 Poster_

### Official Review · Reviewer_xcHp · 2025-10-31

**Soundness:** 3
**Presentation:** 3
**Contribution:** 2
**Rating:** 4
**Confidence:** 4

**Summary:**

Offline GCRL methods trained with sparse rewards struggles in especially long-horizon tasks, since learned value functions tend to be noisy. Task specific hand-crafted reward functions can address this problem, but the task-based reward design is challenging and not practical. It is stated that previous work mostly focuses on online GCRL and recent offline GCRL approaches employ graph based distance classifiers, which fail to scale to large datasets. In this paper, considering the importance of scalability and generalization, an occupancy measure based approach is proposed for offline GCRL. The main idea is that the occupancy measure over future states is learned via flow matching, and then a reward function is defined to guide the policy via occupancy measure matching. Overall the idea is interesting and experimental results show such an occupancy measure based algorithm improves the performance of offline GCRL in long-horizon settings.

**Strengths:**

- **Interesting Approach**: Learning the occupancy measure via flow matching and then using this similarity as the reward function is a novel and interesting approach.
- **Improved Non-monotonicity**: It is nicely presented that the proposed approach has lower non-monotonicity when compared to sparse rewards.
- **Analysis of the Proposed Approach**: The experimental analyses are well-presented, clearly showing the effectiveness of ORS over sparse rewards.

**Weaknesses:**

- **Computational Overhead**: The computational overhead of the proposed approach is not discussed. ORS requires learning an occupancy measure before training the policy, making it appear computationally more complex than baselines. Therefore, the computational overhead should be discussed and compared with both graph-based and non-graph-based offline GCRL baselines.
- **Novelty**: Occupancy measure matching has already been employed by recent works for GCRL. The contribution over these methods is not discussed, which makes the novelty questionable. The benefits of ORS over recent literature (a-b) must be elaborated.
- **Baselines**: In the experiments, recent relevant works (a-c) are omitted from the comparisons.
- **Experimental Setting**: The experiments only cover long-horizon tasks. The applicability of the proposed method to short- and medium-horizon tasks should be discussed.
- **Experimental Results**: The environments used between Table 1 and Table 2 are not the same. It is not clear why some environments are included in Table 1 but omitted in Table 2. For a clear and fair evaluation of ORS, all environments in Table 1 must also be included in Table 2.
- **Vague Explanation**: The main assumption of the paper (in long-horizon tasks, the value function exhibits a high level of non-monotonicity) is evaluated under section 3.2, however it is not clear how $\hat{V}(s,g)$ is trained. It is not explained how the authors obtained $\hat{V}(s,g)$ in 3.2, or which algorithm was used and how $\hat{V}(s,g)$ was trained.

[a]: Sikchi, Harshit, et al. "Score models for offline goal-conditioned reinforcement learning." The Twelfth International Conference on Learning Representations. 2023.

[b]: Ma, Jason Yecheng, et al. "Offline goal-conditioned reinforcement learning via $ f $-advantage regression." Advances in neural information processing systems 35 (2022): 310-323.

[c]: Zhou, John Luoyu, and Jonathan C. Kao. "Flattening Hierarchies with Policy Bootstrapping." Workshop on Reinforcement Learning Beyond Rewards@ Reinforcement Learning Conference 2025.

**Questions:**

- How computationally complex is the proposed method? Can you please provide computational efficiency comparisons with baselines and graph-based solutions?
- Can you please elaborate on the benefits of the proposed approach over recent literature [a-b]?
- Can you please compare the proposed method with [a-c], in addition to the current baselines?
- Can you please elaborate on whether this approach would also be useful in medium or short-horizon tasks?
- Can you please clarify how $\hat{V}(s,g)$) was trained for the analysis in Section 3.2? What algorithm was used?

---

> ### Author Response · Authors · 2025-11-26
> **Response to Reviewer xcHp**
>
> We thank the reviewer for their detailed review. In our response, we provide detailed responses to each of weaknesses highlighted in your review: computational overhead, novelty, baselines, expt. setting, Table 2 results and $\hat{V}(s,g)$. Please find our response below:
>
> ### 1. Computational Overhead:
>
> The average wall-clock time is as follows:
>
> | Method | Component | Time per iteration (ms) |
> | :--- | :--- | :---: |
> | **ORS** | Occupancy model | 14 |
> | | Reward function | 8 |
> | | Policy training | 6.5 |
> | **GO-FRESH** | R-NET training | 9.5 |
> | | Policy training | 7.2 |
>
> GO-FRESH also has a graph computation time that ranges between 1.5-10 mins depending on the task and dataset. For default GCIQL, policy training takes 7.2 ms per iteration. All times were reported on a single NVIDIA RTX A6000 GPU. We typically see that reward function and occupancy model losses converge in around 500k iterations. R-Net training typically runs for 500k iterations.
>
> While ORS does lead to a slightly higher computational overhead, it also leads to a 2.3 $\times$ increase in performance over GCIQL, its base algorithm, and exhibits stronger performance than GO-FRESH. GO-FRESH requires training a distance classifier, training a graph memory and building a graph. This involves: tuning for the size of the R-Net, number of edges in the graph, number of negative pairs, distance threshold for the R-Net and careful weighing of local and global rewards during policy training. In contrast, ORS is simple to train and to tune and performs effective, theoretically motivated credit assignment without heuristics or graph building. We agree that increasing the efficiency of ORS is an important direction for future work.
>
> ### 2.Novelty:
>
> We provide a more detailed discussion of the differences between ORS and occupancy-matching-based methods such as GoFAR and SMORe are required and will add these to the main paper:
>
> GoFAR and SMORe pose offline RL as an occupancy matching problem and then solve for its dual formulation **without directly learning an occupancy measure**. They either learn a discriminator or a score function as a proxy for the value (SMORe calls this a dual value function) leading to a complex constrained optimization objective (**Eq. 8 of the SMORe paper**) which, as we see in practice, could be difficult to optimize and leads to poor performance (as evidenced SMORe results below).
>
> ORS follows a completely different approach: **we directly learn the dataset occupancy measure** from offline data using a flow-matching generative model and extract a reward function that encodes goal reaching information using a simple objective. **ORS does not solve for the dual RL problem directly and is compatible with any typical actor-critic algorithm**.
>
> We include an empirical comparison to SMORe as part of this response (further below) and the results show that ORS is far more effective than SMORe on a variety of tasks. ***Could you please let us know if this discussion and the experiments below adequately answer your questions about novelty?***
>
> ### 3.Comparison with occupancy-matching methods and SAW:
>
> As suggested, we include comparisons to SAW. As SMORe has strictly better performance than GO-FAR (Tables 1 and 2 of the SMORe paper), we add it as a representative dual-RL baseline and compare ORS to SMORe:
>
> | Dataset | SAW | SMORe | ORS (ours) |
> | :--- | :---: | :---: | :---: |
> | antmaze-large-navigate | 86 ± 5 | 22 ± 5 | **88 ± 7** |
> | antmaze-giant-navigate | 48 ± 9 | 1 ± 1 | **56 ± 9** |
> | antmaze-large-explore | **15 ± 8** | 0 ± 0 | **22 ± 7** |
> | cube-double-play | **40 ± 7** | 2 ± 2 | **45 ± 7** |
> | cube-triple-play | 6 ± 6 | 0 ± 0 | **37 ± 8** |
> | puzzle-4x4-play | 17 ± 12 | 0 ± 0 | **70 ± 5** |
> | puzzle-4x5-play | 8 ± 4 | 0 ± 0 | **20 ± 0** |
> | puzzle-4x6-play | 8 ± 4 | 0 ± 0 | **20 ± 2** |
> | cube-triple-noisy | 0 ± 0 | 1 ± 1 | **22 ± 7** |
> | puzzle-4x4-noisy | 3 ± 3 | 0 ± 0 | **56 ± 7** |
> | puzzle-4x6-noisy | 8 ± 6 | 0 ± 0 | **19 ± 1** |
> | scene-play | 63 ± 6 | 8 ± 2 | **80 ± 4** |
> | scene-noisy | 33 ± 6 | 3 ± 2 | **40 ± 5** |
>
> We see that ORS exhibits better performance overall than SAW and outperforms SMORe, which performs poorly in comparison, on a variety of tasks on OGBench.
>
>
> ### 4. Short and Medium Horizon Tasks:
>
> We had ensured that **our analysis in Tables 1 and 2 already cover medium horizon tasks**. These include **cube-double** (500 steps) and **puzzle 4x4** (500 steps). Please refer to **Table 7 of the OGBench paper** for a list of tasks and their horizons. As per your suggestion, we also show results on a short horizon task, cube-single-play (200 steps):
>
> | Dataset | GCBC | GCIVL | GCIQL | QRL | CRL | HIQL | ORS (ours) |
> | :--- | :---: | :---: | :---: | :---: | :---: | :---: | :---: |
> | cube-single-play | 6 ± 2 | 53 ± 4 | 68 ± 6 | 5 ± 1 | 19 ± 2 | 15 ± 3 | **76 ± 7** |

---

> > ### Author Response · Authors · 2025-11-26
> >
> > ### 5. Completeness of experimental results in Table 2:
> >
> > We apologise for the confusion. We clarify that Table 2 only contains results on the difficult long-horizon tasks from OGBench, and compares ORS with strategies specifically designed for addressing long horizons. We provide a revised Table 2 below with results on all tasks, as you suggested:
> >
> > | Dataset | HIQL | n-step GCIQL | n-step GCIQL-OTA | Go-Fresh | ORS (ours) |
> > | :--- | :---: | :---: | :---: | :---: | :---: |
> > | antmaze-large-navigate | **91 ± 2** | 53 ± 9 | 90 ± 4 | 88 ± 3 | 88 ± 7 |
> > | antmaze-giant-navigate | **72 ± 7** | 1 ± 1 | 26 ± 5 | 30 ± 10 | 56 ± 9 |
> > | antmaze-large-explore | 0 ± 0 | 0 ± 0 | 0 ± 0 | **38 ± 10** | 22 ± 7 |
> > | cube-double-play | 6 ± 2 | 4 ± 3 | 3 ± 2 | 17 ± 6 | **45 ± 7** |
> > | cube-triple-play | 3 ± 2 | 1 ± 1 | 2 ± 2 | 18 ± 5 | **37 ± 8** |
> > | puzzle-4x4-play | 7 ± 2 | 46 ± 5 | **85 ± 4** | 74 ± 6 | 70 ± 5 |
> > | puzzle-4x5-play | 8 ± 4 | 5 ± 2 | 19 ± 1 | **20 ± 1** | **20 ± 0** |
> > | puzzle-4x6-play | 3 ± 1 | 14 ± 3 | 15 ± 3 | 17 ± 4 | **20 ± 2** |
> > | cube-triple-noisy | 2 ± 1 | 2 ± 1 | 2 ± 1 | 5 ± 4 | **22 ± 7** |
> > | puzzle-4x4-noisy | 3 ± 3 | 0 ± 0 | 0 ± 0 | **50 ± 5** | **56 ± 7** |
> > | puzzle-4x6-noisy | 2 ± 1 | 12 ± 6 | 15 ± 3 | **19 ± 4** | **19 ± 1** |
> > | scene-play | 38 ± 3 | 26 ± 7 | 42 ± 7 | 56 ± 10 | **80 ± 4** |
> > | scene-noisy | 25 ± 4 | 2 ± 2 | 3 ± 2 | 34 ± 5 | **40 ± 5** |
> > | **Mean** | 20.0 | 12.7 | 23.2 | 35.8 | **44.2** |
> >
> > We emphasize again that ORS is complementary and can be applied with any of the algorithms above, without making any changes to the algorithm.
> >
> > ### 6. Computation of $\hat{V}(s,g)$:
> >
> > In **Sec. 3.2 (Lines 187-198)**, we mention that $\hat{V}(s,g)$ is computed analytically along expert trajectories, i.e., it is computed as the discounted sum of rewards for that trajectory. We also mention how we inject noise into $\hat{V}(s,g)$ for the purpose of analysis in **lines 197-198**. We will make this clearer in the final version of the paper.

---

### Official Review · Reviewer_bbdp · 2025-11-01

**Soundness:** 2
**Presentation:** 2
**Contribution:** 2
**Rating:** 4
**Confidence:** 2

**Summary:**

This paper addresses the problem of insufficient signals in offline goal-conditioned reinforcement learning (Offline Goal-Conditioned RL, GCRL) under sparse rewards and long-horizon tasks. It proposes Occupancy Reward Shaping (ORS), a learning-based reward shaping method grounded in the occupancy measure, which can capture temporal dependencies in long-horizon tasks. By integrating flow matching for fitting, ORS distills goal achievement information from the occupancy measure into a generalizable reward function.

**Strengths:**

1. The paper is clearly written, logically structured and well organized. Technical details are thoroughly presented.
2.The theoretical analysis is solid, providing proofs of convergence and an analysis of reward monotonicity. Starting from empirical evidence that sparse rewards lead to non-monotonic value functions, the paper proposes a reward shaping idea based on the occupancy measure, forming a complete logical chain.
3.Experiments cover both locomotion and manipulation tasks. The experimental design is rigorous, and the results consistently demonstrate strong performance.
4.The approach is compatible with existing offline goal-conditioned RL algorithms, and the analysis of the value function non-monotonicity is insightful. Experiments validate that ORS effectively alleviates this issue.

**Weaknesses:**

1.While multiple tasks from OGBench are used, all of them are simulated environments. There is no mention of testing on real-world data or environments with different physical properties. It remains unclear whether the algorithm remains stable under real-world physics or diverse visual conditions.
2.The theoretical assumptions are relatively strong. Discussions or brief experiments demonstrating robustness under stochastic dynamics are needed, or clarification on whether these assumptions still hold approximately in practice.
3.Ablation studies are insufficient. Although the text mentions “conduct detailed analyses and ablations,” the contribution of different ORS components to the final performance is not shown.

**Questions:**

1.The κ parameter in the ablation is highly sensitive. Could adaptive scheduling of κ improve stability?
2.Does the optimality guarantee in Theorem 1 depend on data coverage quality? In datasets covering only part of the optimal path, can ORS still maintain optimality?

---

> ### Author Response · Authors · 2025-11-26
> **Response to Reviewer bbdp**
>
> Thank you for your valuable feedback on our paper. To address your concern regarding real-world datasets with different physical properties, we evaluate ORS on 3 Tokamak (nuclear fusion reactor) control tasks with real offline data. We understand your concerns about theoretical assumptions, ablations, varying $\kappa$ and suboptimal data and provide detailed answers below:
>
> ## **Weaknesses**:
>
> ### 1. Real-world datasets:
> We perform experiments on challenging real world datasets from a nuclear fusion reactor called a Tokamak. This involves controlling plasma evolution: a complex physical system with highly stochastic and non-linear dynamics. Please refer to the **combined response** titled “**Evaluation on Real-World Datasets**” for a detailed description of tasks and results. ORS consistently performs the best and exhibits stable performance. ***Could you please let us know if this addresses your specific concern***?
>
> ### 2. Concerns about theoretical assumptions:
>
> Assumption A.1 (Deterministic Dynamics) is a common simplifying assumption used to ensure the analytical tractability of our proofs, allowing us to clearly derive the properties of ORS  For instance, [1] also rely on deterministic dynamics. However, we clarify that our theoretical insights hold approximately under stochastic dynamics: In physical systems satisfying Lipschitz continuity (a common assumption), the value error introduced by stochastic transitions is bounded. Consequently, the value function retains its global structure in expectation and the learned policy receives a consistent and valid gradient pointing towards goal.
>
> This is validated empirically by the strong performance of ORS on Tokamak control tasks. Ablations with noisy goals to the reward model (**answer to Question 1 in response to reviewer noM7**) further show robustness to stochasticity. ***Having provided a discussion and experiments showing robustness to stochasticity, could you please let us know if we addressed your questions***?
>
> ### 3. Insufficient Ablations:
>
> The main component of ORS is the reward model that learns rewards extracted from the occupancy model. We mention that we “conduct detailed analyses and ablations” owing to the following experiments in our paper:
>
> 1. We analyse the non-monotonicity in the value function learnt by ORS rewards vs sparse rewards over varying noise levels, plot value functions and visualize learned rewards : **Figures 3, 4, 5 and Lines 402-410**.
> 2. We ablate the choice of reward function, the core focus of our paper: **Lines 441 to 453 and Figure 6**. Furthermore, in the **answer to Question 2 of reviewer noM7**, we evaluate the performance of L2 rewards and show its ineffectiveness.
> 3. We ablate over different values of $\kappa$ (**Lines 454 to 461 and Table 3**).
> 4. We perform experiments with noisy goals in our **answer to Question 1 of reviewer noM7**.
> 5. We also ablate over adaptive scheduling of the expectile parameter and show the results below.
>
> ***We are happy to provide results on additional ablations, if you could let us know which specific component requires further ablation studies***.
>
> ## **Questions**:
>
> ### 1. Sensitivity to $\kappa$:
>
> As shown in **Fig. 3 of [2]**, the variation in performance with $\kappa$ is expected and by design of IQL. Furthermore, we conduct experiments on scene-play with a linear schedule of kappa with up-down representing starting with a value of 0.9 and decaying it to 0.6 over 500k steps, and down-up representing the opposite of this:
>
> | Dataset | ORS down-up | ORS up-down | ORS |
> | :--- | :---: | :---: | :---: |
> | scene-play | 70 ± 15 | 72 ± 5 | **80 ± 4** |
>
> ### 2. Suboptimal data:
>
> As shown by our strong results (**Tables 1 and 2**) on cube-triple-noisy, scene-noisy, puzzle-4x4-noisy, antmaze-large-explore and puzzle-4x6-noisy, ORS remains effective even with highly sub-optimal data. From a theoretical perspective, as discussed in [3], theoretical guarantees for offline RL fundamentally face a trade-off between assumptions on function classes (e.g., Bellman-completeness) and data coverage (e.g., concentrability). Relaxing both simultaneously is known to be theoretically intractable. Since ORS does not impose restrictive assumptions on the function class (which are often difficult to satisfy in practice), relying on data coverage assumptions is necessary and standard.
>
> **References**
>
> [1] Kristian Hartikainen, Xinyang Geng, Tuomas Haarnoja, and Sergey Levine. Dynamical distance
> learning for semi-supervised and unsupervised skill discovery. International Conference on Learning Representations (ICLR), 2020
>
> [2] Kostrikov, Ilya, Ashvin Nair, and Sergey Levine. "Offline reinforcement learning with implicit q-learning." arXiv preprint arXiv:2110.06169 (2021)
>
> [3] Wenhao Zhan, Baihe Huang, Audrey Huang, Nan Jiang, and Jason Lee. Offline reinforcement
> learning with realizability and single-policy concentrability. In Conference on Learning Theory,
> pp. 2730–2775. PMLR, 2022

---

### Official Review · Reviewer_noM7 · 2025-11-08

**Soundness:** 3
**Presentation:** 2
**Contribution:** 2
**Rating:** 4
**Confidence:** 3

**Summary:**

The paper proposes to address the struggles of offline goal-conditioned RL on long horizon tasks via reward shaping. In particular, it propposes a novel reward shaping method, occupancy reward shaping, trained using flow matching to perform effective credit assignment as a reward function. Experiments show that it improves over prior Offline GCRL methods on long-horizon lcomotion and manipulation tasks in simulation.

**Strengths:**

- clear motivation and extensive discussion on background
- provides proofs for theoretical guiarantee

**Weaknesses:**

- It would be more convincing if results can also be demonstrated on real-world robotics tasks, where both data quantity and quality are lower
- There should be more discussion on other ways of computing dense reward information
- Results seem only to be marginally better, most of the gains over GO-FRESH are on 2 tasks
- how did the authors select the tasks in the benchmark? why are tasks like ant soccer and humanoid maze not selected?

**Questions:**

- What is the quality and quantity of data that is required to train such a reward model? Will there be circumstances where the reward model is not accurate? If so how is the performance affected
- In an offline RL setting, since there is no online interaction, why can't the reward be simply obtained using distance of current state to goal?
- Can you discuss the comparison of your approach against works like GoFar (Ma et al), where shaped reward is not used, and goal reaching behavior is direclty learned by minimizing divergence between policy and expert's  goal conditioned state occupancy?

---

> ### Author Response · Authors · 2025-11-26
> **Response to Reviewer noM7**
>
> We thank the reviewer for their feedback. We understand and acknowledge your main concern about evaluation on real-world tasks. To address this, we provide results on 3 real-world tasks. Please find our detailed response to all weaknesses and questions below:
>
> ## **Weaknesses**:
>
> ### 1. Real-World Datasets:
>
> We provide results on real-world datasets in the **combined response** above, titled “**Evaluation on Real-World Datasets**”. ***Could you please specify if these experiments address your specific concerns?***
>
> ### 2. Discussion on Reward Shaping:
>
> We provide a detailed discussion on this topic in the sub-section “Reward Shaping” of **Section 2 (Lines 112–128)** covering different approaches to reward shaping in offline and online RL. ***Kindly let us know what other details you think might be relevant to discuss here; we would be happy to revise the paper further.***
>
> ### 3. Results seem only to be marginally better, most of the gains over GO-FRESH are on 2 tasks
>
> We understood this concern to refer to a requirement for evaluating the statistical significance of ORS in comparison with GO-FRESH. We address this by showing that ORS is significantly better than GO-FRESH on the majority of tasks, over varying levels of task difficulty, dataset size, and dataset quality, as shown below:
>
> | Dataset | p-value | Significantly higher/lower ($\uparrow/\downarrow$) | ORS | Go-Fresh |
> | :--- | :--- | :---: | :---: | :---: |
> | Antmaze-large-navigate | $0.83$ | - | $88 \pm 7$ | $88 \pm 3$ |
> | Antmaze-giant-navigate | $2\times10^{-6}$ | $\uparrow$ | $56 \pm 9$ | $30 \pm 10$ |
> | Cube-double-play | $5\times10^{-8}$ | $\uparrow$ | $45 \pm 7$ | $17 \pm 6$ |
> | Puzzle-4x4-play | $0.045$ | $\downarrow$ | $70 \pm 5$ | $74 \pm 6$ |
> | Cube-triple-play | $9\times10^{-6}$ | $\uparrow$ | $37 \pm 8$ | $18 \pm 5$ |
> | Puzzle-4x5-play | $0.56$ | - | $20 \pm 0$ | $20 \pm 1$ |
> | Puzzle-4x6-play | $0.044$ | $\uparrow$ | $20 \pm 2$ | $17 \pm 4$ |
> | Antmaze-large-explore | $9\times10^{-4}$ | $\downarrow$ | $22 \pm 7$ | $38 \pm 10$ |
> | Puzzle-4x4-noisy | $0.025$ | $\uparrow$ | $56 \pm 7$ | $50 \pm 5$ |
> | Scene-play | $8.2\times10^{-6}$ | $\uparrow$ | $80 \pm 4$ | $56 \pm 10$ |
> | Scene-noisy | $4.4\times10^{-4}$ | $\uparrow$ | $40 \pm 5$ | $34 \pm 5$ |
> | Cube-triple-noisy | $0.0026$ | $\uparrow$ | $22 \pm 7$ | $5 \pm 4$ |
> | Puzzle-4x6-noisy | $0.64$ | - | $19 \pm 1$ | $19 \pm 4$ |
>
> ***Could you please let us know if the statistically significant performance improvements address your concerns or specify what level of improvement would be required to mark a clear advancement?*** If we have misunderstood this concern, please let us know; we will provide further revisions to address it.
>
> ### 4. Task selection
>
> OGBench tasks are broadly categorized into 4 types: maze navigation, cube manipulation, puzzle manipulation, and scene tasks. We selected tasks to be representative of these 4 categories. Within each category, we include tasks of varying complexity (e.g., cube-double and cube-triple; antmaze-large and antmaze-giant; puzzle-4x4, puzzle 4x5, and puzzle 4x6), and varying dataset quality as mentioned in **Lines 343-345** of our paper.
>
>
> ## **Questions**:
>
>
> ### 1. Quality and quantity of data required to train reward model
>
> Please refer to **Table 8 of [1]** for a list of dataset sizes for each task in **Tables 1 and 2** of our paper. **Tables 1 and 2** of our paper show evaluation results on sub-optimal datasets for locomotion and manipulation: scene-noisy, cube-triple-noisy, puzzle-4x4-noisy, puzzle-4x6-noisy and antmaze-large-explore. In theory, given enough samples, the reward model will accurately estimate the true reward as shown by **Proposition 2 (Line 285)**. In practice, we see that the reward model estimates rewards that produce state-of-the-art results on datasets of different sizes and varying data quality. We studied this further by adding a 10\% gaussian noise to the goals during reward model distillation (based on **Eq. 5 in Line 296** of our paper) and using this reward model to train a policy for the scene-play task. The performance was **80 $\pm$ 6** as opposed to **80 $\pm$ 4** without adding noise. We therefore did not observe a degradation in performance.
>
>
> ### 2. Reward as distance of current state to goal:
>
> We performed further ablations using a reward computed as you mentioned: i.e., the L2 distance between the current state and goal. Our experiments on cube-triple-play and antmaze-giant-navigate show that L2 rewards are ineffective:
>
> | Dataset | L2 rewards | ORS |
> | :--- | :---: | :---: |
> | antmaze-giant-navigate | 3 ± 2 | **56 ± 9** |
> | cube-triple-play | 3 ± 1 | **37 ± 8** |
>
> ### 2. Comparison with Go-FAR:
>
> Kindly refer to our ***response to reviewer xcHp*** for a comparison with occupancy-matching-based algorithms .
>
> **References**
>
> [1] Park, Seohong, Kevin Frans, Benjamin Eysenbach, and Sergey Levine. "Ogbench: Benchmarking offline goal-conditioned rl." arXiv preprint arXiv:2410.20092 (2024).

---

### Author Response · Authors · 2025-11-26
**Evaluation on Real-World Datasets**

Since both **reviewers noM7 and bbdp (reviews 1 and 2)** suggested evaluation on real-world datasets, we provide a combined response below:

We would like to emphasize that OGBench [1] is widely used in offline GCRL literature and represents the state-of-the-art for benchmarking offline GCRL. However, we agree that real-world results are important. Taking into account reviews 1 and 2, we perform evaluation on real-world datasets [2] for actuator control in a nuclear fusion reactor called a Tokamak.

We use a dataset of raw sensor and actuator data collected from the DIII-D tokamak located in San Diego, CA, USA. Each trajectory corresponds to one plasma discharge, with a control frequency of 25 ms. A control task corresponds to tracking certain state variables of the plasma. While we were not able to get experiment time on the reactor on short notice, we evaluated each RL algorithm using rollouts from a thoroughly tested and accurately learned model of plasma dynamics [2] as a proxy. The reward at each time-step during evaluation corresponds to the L2 distance between the goal variables to track and the actual achieved quantities. We evaluated each algorithm over 10 goals. The offline dataset size was roughly 50k samples after data cleaning and processing. The state and action spaces are:

| Category | Variables |
|---------|-----------|
| State   | **Scalar states:** $\beta_N$, Internal Inductance, Line-avg Density, Loop Voltage, Stored Energy. **Profile states:** Electron Density, Electron Temp, Pressure, Safety Factor, Ion Temp, Ion Rotation. |
| Action  | Power Injected, Torque Injected, Total Deuterium Gas Injection, Total ECH Power |

Please refer to [2, 3, 4] for detailed explanations regarding variables and actuators. We evaluated over 3 tasks: $\beta_N$ tracking, Electron density tracking and Ion rotation tracking. $\beta_N$ is the normalized ratio between plasma pressure  and magnetic pressure, a key quantity serving as a rough economic indicator of efficiency. Ion rotation is crucial to maintaining stable plasma and magnetic confinement. Electron density crucially influences plasma stability and overall power output of the tokamak. Overall, tokamak control involves highly non-linear, stochastic dynamics and complex physics and even small improvements in control performance signify substantial advancement. The results are as follows:

| Task            | GCBC             | CRL              | HIQL             | GCIQL             | n-step GCIQL-OTA | GO-FRESH          | SMORe            | ORS              |
|-----------------|------------------|------------------|------------------|-------------------|------------------|-------------------|------------------|------------------|
| Tokamak $\beta_N$  | **-48.92 ± 5.74** | -59.91 ± 22.86  | **-45.26 ± 3.83** | **-46.42 ± 11.63** | -67.73 ± 9.73    | **-48.20 ± 11.72** | -56.06 ± 12.71   | **-44.76 ± 8.42** |
| Tokamak density | -41.6 ± 3.6      | -44.2 ± 2.2      | -49.8 ± 7.5      | **-30.7 ± 12.1**  | **-27.1 ± 2.5**  | -38.6 ± 12.6      | -54.0 ± 10.8     | **-26.9 ± 6.6**   |
| Tokamak rotation| -27.4 ± 5.5      | -27.3 ± 3.1      | -24.4 ± 1.3      | -25.8 ± 6.7       | -30.1 ± 6.9      | -28.9 ± 7.4       | -28.5 ± 6.7      | **-22.4 ± 0.6**   |

ORS consistently achieves best performance on these real-world datasets, highlighting its effectiveness.

**References**

[1] Park, Seohong, Kevin Frans, Benjamin Eysenbach, and Sergey Levine. "Ogbench: Benchmarking offline goal-conditioned rl." arXiv preprint arXiv:2410.20092 (2024).

[2] Char, Ian, Youngseog Chung, Joseph Abbate, Egemen Kolemen, and Jeff Schneider. "Full shot predictions for the diii-d tokamak via deep recurrent networks." arXiv preprint arXiv:2404.12416 (2024).

[3] Char, I., J. Abbate, L. Bardoczi, et al. Offline model-based reinforcement learning for tokamak 413 control. In Learning for Dynamics and Control Conference, vol. 211, pages 1357–1372. 2023.

[4] Abbate, Joseph, Rory Conlin, and Egemen Kolemen. "Data-driven profile prediction for DIII-D." Nuclear Fusion 61, no. 4 (2021): 046027.

---

### Author Response · Authors · 2025-12-02
**Discussion Summary**

Dear AC,

We thank you for your time and effort in reviewing our submission.

Our work proposes a novel method for reward shaping to improve credit assignment in offline goal-conditioned RL. We derived theoretical guarantees and demonstrate strong improvements over the state-of-the-art in 14 standard simulated tasks and 3 real-world tasks. During the discussion phase, we have made an effort to address every weakness and question posed by each reviewer.

To do this, we ran ORS vs. baselines on 3 new real-world tasks, performed 3 new sets of ablations (noisy goals, L2-rewards and $\kappa$ scheduling), compared ORS with 2 new baselines over 13 tasks and added comparisons with HIQL, n-step GCIQL and n-step GCIQL-OTA on 6 more tasks. Furthermore, we provided detailed responses to all other technical concerns and questions.

More specifically:

--------
**Major concern regarding real-world results**:

Reviewers **noM7** and **bbdp** suggested evaluation on real-world datasets to strengthen the convincingness of empirical analysis. To address this concern, we evaluate ORS on 3 real-world Tokamak control tasks and show that ORS performs best on all 3 tasks, demonstrating that it is effective in real-world tasks. Please find the detailed response in our response titled **Evaluation on Real-World Datasets**.

--------
**Reviewer noM7**:

In our response titled **Response to reviewer noM7**, we provide real-world results (**weakness 1**) and show that ORS is significantly better than GO-FRESH on the majority of OGBench tasks (**weakness 3**). We address the reviewer's concerns regarding discussion on reward shaping (**weakness 2**) and task selection (**weakness 4**). We also address the reviewer's questions regarding performance of ORS under varying dataset size and quality (**question 1**), using L2-distance as reward (**question 2**) and comparison with occupancy-matching methods (**question 3**).

--------
**Reviewer bbdp**:

In our response titled **Response to reviewer bbdp**, we provide real-world results (**weakness 1**) and positively address concerns regarding theoretical assumptions with a discussion and empirical results (**weakness 2**). We provide a detailed list to clarify the extent of our analyses and ablations (**weakness 3**). We perform further ablations to test our sensitivity to $\kappa$ (**question 1**) and discuss how ORS performs under suboptimal data (**question 2**).

--------
**Reviewer xcHp**:

We note that the questions of the reviewer correspond to each of the weaknesses. To address the reviewer's concerns, in our response titled **Response to reviewer xcHp**, we provide a discussion on computational overhead (**weakness 1**), show that ORS is novel and markedly different from occupancy-matching methods (**weakness 2**) and further show that ORS performs much better than occupancy-matching methods and SAW, the method from ***Flattening Hierarchies with Policy Bootstrapping*** (**weakness 3**).

We clarify that our empirical analysis already includes medium-horizon tasks and show the ORS performs best on the short-horizon task  **cube-single-play** (**weakness 4**). We perform experiments to address the reviewer's concerns regarding completeness of Table 2 (**weakness 5**) and clarify that we provide details on computing $\hat{V}(s,g)$ in **Sec. 3.2 (Lines 187-198)** (**weakness 6**).

----------------

We sincerely thank the reviewers for their reviews. While we did not get responses from reviewers with an updated assessment of our paper; we hope that the AC will consider the extent to which our responses have strengthened our paper during evaluation. We refer the AC to the full discussion below for further clarifications and details.



Sincerely,

The Authors.

---

### Meta-Review · Area_Chair_uwzB · 2026-01-07

**Summary:**

**Summary of the paper**

This paper studies offline goal-conditioned RL (GCRL) under sparse rewards, with a focus on long-horizon tasks where credit assignment becomes difficult and learned value functions can become noisy/non-monotonic. The authors propose Occupancy Reward Shaping (ORS), which first learns a dataset occupancy measure via flow matching, then distills goal-reaching information from that occupancy model into a goal-conditioned shaped reward based on a Wasserstein-2 formulation. The shaped reward is then plugged into standard offline GCRL actor-critic training (the paper emphasizes compatibility with existing methods). Experiments on OGBench locomotion (AntMaze) and manipulation (Cube/Puzzle/Scene) tasks show ORS outperforming the base sparse-reward method and improving over prior offline GCRL reward-shaping baselines on many long-horizon settings.

**Summary of the reviewers' concern**

All three reviewers gave a 4/10 (“marginally below acceptance”) rating, primarily due to concerns about (i) evaluation scope and (ii) positioning/experimental completeness rather than a belief that the core idea is invalid. Reviewer noM7 emphasized that the work would be more convincing with real-world robotics evidence, questioned whether gains over GO-FRESH were “marginal,” asked about data requirements for training the reward model, and asked for discussion/comparisons to other ways of deriving dense signals (including distance-to-goal style rewards and comparisons to occupancy-matching approaches like GoFar). Reviewer bbdp similarly raised the absence of real-world evaluation, noted that the theoretical assumptions (e.g., deterministic dynamics) seemed strong and asked for robustness discussion/experiments under stochasticity and data limitations, and also stated that ablations were insufficient (including sensitivity of the (\kappa) parameter and questions about how guarantees depend on coverage). Reviewer xcHp focused on computational overhead, novelty/positioning relative to prior occupancy-matching methods, missing recent baselines, clarity on the experimental setting (including why some environments appear in one table but not another), applicability beyond long-horizon tasks, and one methodological clarity issue about how a key quantity in the value non-monotonicity analysis was obtained/trained.

**Reviewer Concerns:**

I belive that the following concerns are more or less completely addressed by a detailed and thorough rebuttal by the authors.

1. Real-world evaluation (Reviewers noM7 & bbdp): The authors explicitly added/reported real-world offline-data experiments (Tokamak/nuclear fusion reactor control tasks)
2. "Marginal gains over GO-FRESH"  (Reviewer noM7): The rebuttal provided a statistical significance analysis (p-values) comparing ORS vs GO-FRESH across multiple tasks/difficulty/data-quality settings, arguing improvements are significant on a majority of tasks.
3. Ablations (Reviewer bbdp): The rebuttal pointed to specific analyses already in the paper (value non-monotonicity analyses, reward-function ablations, (\kappa) sweeps) and also added an adaptive scheduling experiment for (\kappa).
4. Theoretical assumptions (Reviewer bbdp): The rebuttal clarified the role of the deterministic assumption for tractability and argued the insights hold approximately under stochastic dynamics, citing existing works and empirical robustness on experiments.
5. Computational overhead + novelty/baselines concerns (Reviewer xcHp): The rebuttal provided wall-clock timing breakdowns and argued ORS only has a modest overhead relative to GO-FRESH

I believe that the remaining concerns, if any, are minor and do not affect the quality of the paper.

**Reviewer Scores:**

Each reviewer is very likely to increase his/her current rating (4/10) to marginally above acceptance (6/10)

---

### Decision · Program_Chairs · 2026-01-26

Accept (Poster)